# A specialized bone marrow microenvironment for fetal haematopoiesis

Yang Liu [1,2,7 ✉], Qi Chen [1,3,4,7], Hyun-Woo Jeong [1,7], Bong Ihn Koh[1], Emma C. Watson [1], Cong Xu[1], Martin Stehling[5], Bin Zhou [6] & Ralf H. Adams [1 ✉]

In adult mammalian bone marrow (BM), vascular endothelial cells and perivascular reticular cells control the function of haematopoietic stem and progenitor cells (HSPCs). During fetal development, the mechanisms regulating the de novo haematopoietic cell colonization of BM remain largely unknown. Here, we show that fetal and adult BM exhibit fundamental differences in cellular composition and molecular interactions by single cell RNA sequencing. While fetal femur is largely devoid of leptin receptor-expressing cells, arterial endothelial cells (AECs) provide Wnt ligand to control the initial HSPC expansion. Haematopoietic stem cells and c-Kit$^+$ HSPCs are reduced when Wnt secretion by AECs is genetically blocked. We identify Wnt2 as AEC-derived signal that activates β-catenin-dependent proliferation of fetal HSPCs. Treatment of HSPCs with Wnt2 promotes their proliferation and improves engraftment after transplantation. Our work reveals a fundamental switch in the cellular organization and molecular regulation of BM niches in the embryonic and adult organism.

[1] Max Planck Institute for Molecular Biomedicine, Department of Tissue Morphogenesis, University of Münster, Faculty of Medicine, Münster D-48149, Germany. [2] School of Medicine, South China University of Technology, University Town Campus, Room 505, Building B2, Guangzhou 510006, China. [3] Center for cell lineage and development, CAS Key Laboratory of Regenerative Biology, Guangdong Provincial Key Laboratory of Stem Cell and Regenerative Medicine, GIBH-HKU Guangdong-Hong Kong Stem Cell and Regenerative Medicine Research Centre, GIBH-CUHK Joint Research Laboratory on Stem Cell and Regenerative Medicine, Guangzhou Institutes of Biomedicine and Health, Chinese Academy of Sciences, Guangzhou 510530, China. [4] China-New Zealand Joint Laboratory on Biomedicine and Health, Guangzhou 510530, China. [5] Flow Cytometry Unit, Max Planck Institute for Molecular Biomedicine, Münster D-48149, Germany. [6] State Key Laboratory of Cell Biology, CAS Center for Excellence in Molecular Cell Science, Shanghai Institute of Biochemistry and Cell Biology, Chinese Academy of Sciences, 320 Yueyang Road, A-2112, Shanghai 200031, China. [7] These authors contributed equally: Yang Liu, Qi Chen, Hyun-Woo Jeong. ✉email: yangl005@scut.edu.cn; ralf.adams@mpi-muenster.mpg.de

Haematopoietic stem cells (HSCs) are a rare cell population characterized by self-renewal capacity and multipotency, which together enable the lifelong generation of all cell types in the haematopoietic system[1]. Transplantation of HSCs after radiotherapy or chemotherapy is a therapeutic treatment for many malignant and non-malignant haematological disorders[2,3]. In the mammalian embryo, HSCs emerge during definitive haematopoiesis from the aortic endothelium[4–7], which is followed by their release into the circulation and transient colonization of the fetal liver[8–10]. HSCs ultimately colonize bone marrow, the primary site for postnatal and adult haematopoiesis[11].

In adult BM, the behavior of haematopoietic stem and progenitor cells (HSPCs) is regulated by complex microenvironments involving multiple cell types, in particular vascular endothelial cells (ECs) and perivascular LepR+ reticular cells[11–17]. Both cell types are a critical source of stem cell factor (SCF; encoded by *Kitl*)[18,19], a cytokine that controls HSC self-renewal and maintenance through the activation of the receptor tyrosine kinase c-Kit. Reticular cells also provide another cytokine, stromal cell-derived factor-1 (Cxcl12)[20], which signals through the G-protein-coupled receptor Cxcr4 and thereby regulates HSPC maintenance, trafficking and homing[21–23].

The vascular microenvironment in bone is also a source of other paracrine acting signals controlling HSC/HSPC function[12], including pleiotrophin (encoded by *Ptn*)[24], vascular endothelial growth factor (VEGF)[25], Notch ligands of both the Jagged and Delta-like subfamily[26–28], the cytokine interleukin-7[29], and the growth factor angiopoietin-1 (encoded by *Angpt1*)[30]. Recent single cell RNA sequencing (scRNA-seq) analysis has provided insight into the properties and heterogeneity of HSPCs and niche-forming cells in adult BM[26,31–36], but it remains unclear whether these findings apply to fetal bone marrow.

Mature hematopoietic cells are also part of the complex BM microenvironment in the adult organism and regulate their own stem cell predecessors, as has been shown for regulatory T-cells[37], certain rare monocytes-macrophages[38], bone-resorbing osteoclasts[39–41], and megakaryocytes[42–44]. While several studies have investigated the transcriptional profile and differentiation of fetal and adult HSPCs at single cell resolution[35,45–47], the properties of the fetal BM stroma and the mechanisms mediating HSC engraftment and expansion in the initial stages of marrow development remain little understood.

In this study, we combine scRNA-seq, flow cytometry, advanced imaging, tissue-specific genetic mouse models and BM transplantation experiments to investigate the interactions between stromal cells and HSPCs in fetal femur. This uncovers that fetal BM is fundamentally different from its adult counterpart with respect to cellular composition and HSPC regulation. Based on our findings, we propose that arterial endothelial cells in the developing BM provide critical signals promoting the de novo colonization of fetal bone by HSPCs.

## Results

### Vascular and HSPC development in fetal BM

To understand the properties and heterogeneity of embryonic HSPCs and niche cells in BM, we characterized the development of femoral vascular network in relation to c-Kit+ haematopoietic cells. Consistent with our previous report[48], formation of femoral vascular network is initiated between embryonic day (E) 14.5 to E15.5 (Supplementary Fig. 1a). The abundance of Endomucin+ (Emcn+) vessels in the primary ossification centre rapidly increases during the following days. This expansion of the bone vasculature involves the specification of AECs, defined by surface markers as CD31+ Emcn− or by high expression of the caveolar protein Caveolin-1, and artery formation is detectable from E15.5 onwards

(Supplementary Fig. 1a–d). Coverage by α-smooth muscle actin+ (αSMA+) vascular smooth muscle cells, a hallmark of larger arteries, is also initiated at E15.5 but is more pronounced at later developmental stages (Supplementary Fig. 1e–g). In contrast, c-Kit+ haematopoietic cells are absent in E15.5 femur (Supplementary Fig. 2a, b), indicating that the formation of a vascular network and artery specification precede the colonization by haematopoietic cells. While c-Kit+ haematopoietic cells are rare in developing BM at E16.5, their number increases rapidly during later embryogenesis (Fig. 1a–c; Supplementary Fig. 2c–f). These c-Kit+ haematopoietic cells are enriched in centre of the diaphysis relative to the metaphysis and endosteum at E18.5 (Supplementary Fig. 2g). Engrafted c-Kit+ haematopoietic cells incorporate EdU (5-ethynyl-2-deoxyuridine), indicating rapid cell division in fetal BM (Supplementary Fig. 2h). Major known HSPC subsets are present in E18.5 femur (Supplementary Fig. 2i).

### Comparative single cell analysis of HSPCs and BM stroma

To compare adult and embryonic HSPCs and niche cells in femur, we isolated BM cells from adult and E18.5 mice (see "Methods"), which is around the earliest time at which femoral haematopoietic cells show long-term repopulating ability[49]. HSPCs and BM stromal cells were enriched by depletion of lineage-positive (Lin+) cells for scRNA-seq analysis (Supplementary Fig. 3a). Unsupervised clustering subdivided the obtained cells into 7 groups, which can be clearly distinguished by the expression of unique markers (Fig. 1d, e; Supplementary Fig. 3b–e). Uniform manifold approximation and projection (UMAP) plots show substantial differences between embryonic and adult samples (Fig. 1d, e), which we investigated further by unsupervised subclustering of embryonic and adult HSPCs and stromal cells.

Although major HSPC subsets are present at E18.5 (Fig. 1f, g; Supplementary Fig. 3f, g), myeloid and megakaryocyte progenitors are less abundant than in adult BM. Moreover, differentially-expressed gene (DEG) analysis at subcluster level shows substantial differences between E18.5 and adult HSPCs with hundreds of DEGs in each subcluster (Fig. 1h), which is consistent with previous reports[35,50,51]. These DEGs include critical transcription factors and receptors mediating HSPC development (Fig. 1i; Supplementary Fig. 3h)[52,53].

The most striking difference between E18.5 and adult samples is visible in the bone marrow stromal cell compartment (defined here as BMSCs) (Fig. 2a). By subclustering of cells, we detect 2 populations of osteolineage cells (OLCs-1, OLCs-2), mitotic BMSCs, adult-enriched BMSCs (aBMSC), and two groups of embryo-enriched BMSCs (eBMSC-1, eBMSC-2) (Fig. 2a, b). *Lepr*+ aBMSCs, representing LepR-expressing reticular niche cells for HSCs, are highly abundant in adult BM (Fig. 2a) but almost completely absent at E18.5 (Fig. 2a)[54–56]. This raises the question whether another BMSC population in fetal BM fulfils the function of LepR+ aBMSCs. The eBMSC-1 population with high expression of collagen III subunit 1 (*Col3a1*high), the proteoglycan decorin (*Dcn*) and the C-type lectin domain family member Gsn/Clec3b, represents the majority (57%) of fetal BMSCs but is substantially reduced in adults (Fig. 2a, b; Supplementary Fig. 4a–c). *Col3a1*high eBMSC-1 cells express typical mesenchymal markers such as collagen I subunit 1 (*Col1a1*) or platelet-derived growth factor receptor α (*Pdgfra*) (Supplementary Fig. 4b).

The paucity of LepR+ reticular cells in fetal BM is confirmed by immunostaining of tissue sections and FACS analysis of genetically labeled cells in *Lepr-Cre R26-mTmG* reporter mice (Fig. 2c; Supplementary Fig. 4e–f; Supplementary Table S1). The distinct molecular properties of embryonic and adult whole BMSCs population are also indicated by the large number of 881

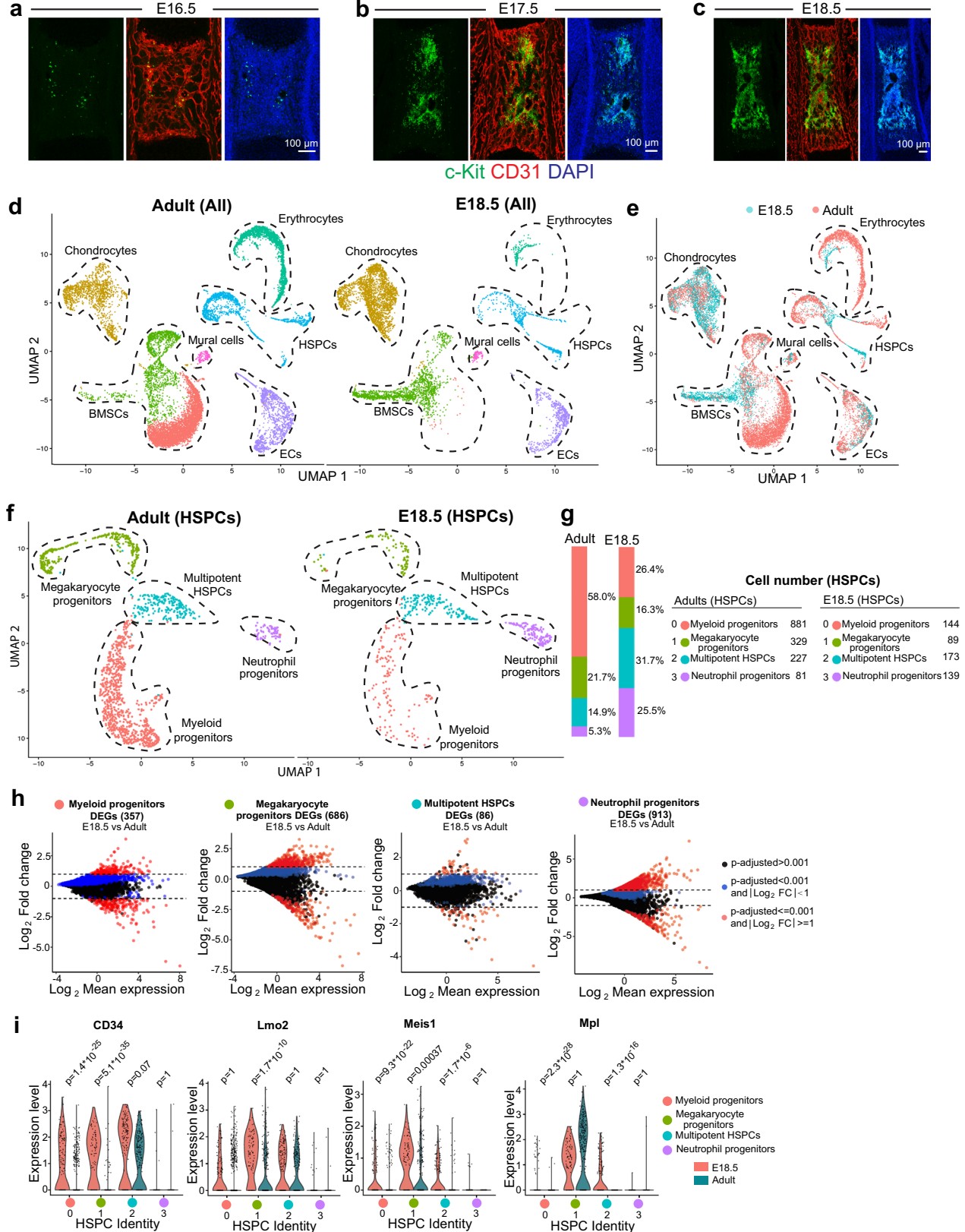

DEGs (Fig. 2d). LepR+ reticular cells are known to provide critical niche factors for HSCs, most prominently stem cell factor (SCF; encoded by *Kitl*), the chemokine Cxcl12 and the growth factor angiopoietin-1 (*Angpt1*)[30]. Consistent with the lack of the LepR+ subpopulation at E18.5, expression of all these transcripts is very low in embryonic BMSCs (Fig. 2e; Supplementary Fig. 4d),

which implies that the secretion of these niche factors by LepR+ cells is not taken over by *Col3a1*+ eBMSCs (Fig. 2e, g). Metacell interrelation analysis (Supplementary Fig. 4g), DEG comparison analysis at the subcluster level (Fig. 2f) and interactome analysis of potential interactions with multipotent HSPCs (Supplementary Fig. 4h) support the differences between adult and fetal BMSC

**Fig. 1 Comparative analysis of embryonic and adult BM components. a–c** Representative overview images showing c-Kit[+] (green) cells in femoral BM at E16.5 (**a**), E17.5 (**b**) and E18.5 (**c**). CD31(Red); Nuclei, DAPI (blue). **d, e** UMAP plots showing unsupervised clustering of BM cells from E18.5 and adult femur after depletion of Lin[+] cells (**d**) and the differences between the sample groups (**e**). **f** UMAP plot showing unsupervised subclustering of E18.5 and adult haematopoietic stem and progenitor cells (HSPCs). **g** Bar chart indicating percentage of cells in each subcluster. Cell numbers are shown on the right. **h** Differentially expressed genes (DEGs) comparing each subcluster of HSPC between E18.5 and adult. Blue dots, $p$-adjusted value < 0.001; Red dots, $p$-adjusted value < 0.001 and Log$_2$ fold change >1. Two-tailed Student's $t$ test. **i** Violin plots showing the expression of selected genes with log normalized values ($y$ axis) for critical transcription factors and receptors in each subcluster of E18.5 and adult HSPCs.

---

populations. Although expression of many known niche factors is low in E18.5 BMSC subclusters, we detect substantially higher expression of another signal acting on HSPCs, the cytokine pleiotrophin (encoded by *Ptn*), in E18.5 BMSCs (Fig. 2g).

Unsupervised subclustering of bone ECs indicates similar major subpopulations at E18.5 and in adults (Fig. 3a). Both groups contain differentiated AECs with low *Emcn* expression and high levels of transcripts for the kinase Bmx, the transcription factor Sox17, the transmembrane ligand ephrin-B2 (*Efnb2*), the gap junction protein connexin 37 (*Gja4*) and Caveolin-1 (*Cav1*) (Fig. 3b; Supplementary Fig. 5a, b). *Stab2*[high] ECs, representing sinusoidal ECs of the BM with important roles in haematopoiesis[26], are present in both E18.5 and adult samples (Fig. 3a). Comparison of embryonic and adult ECs at subcluster level reveals several hundred significant DEGs (Fig. 3c) without overt alterations in secreted niche factors such as *Kitl*, *Cxcl12* or *Pdgfb* (Fig. 3d). Interactome analysis was used to evaluate whether the DEGs in stromal cells and HSPCs are indicative of differences in HSPCs regulation at the transcriptomic level. This analysis suggests substantial changes in potential HSPC-niche cell interactions in embryonic vs. adult bone (Fig. 3e, f). Some of these interactions were validated by previous publications. Examples include fibroblast growth factor 2 (FGF2), which promotes HSPC expansion[57], insulin growth factor 2 (IGF2), a regulator of HSC maintenance[58,59], or tumor necrosis factor ligand superfamily member 9 (TNFSF9), which has been proposed to inhibit stem cell self-renewal[60].

**Initially-engrafted HSCs and HSPCs are associated with AECs.** Next, we asked which cell populations might support the initial engraftment and expansion of HSPCs in fetal BM before the emergence of LepR[+] reticular cells. Our analysis of stained bone sections reveals that CD150[+] CD48[−] CD41[−] Lin[−] HSCs are predominantly aligned with Caveolin-1[+] AECs at E16.5 (Fig. 4a, b; Supplementary Fig. 5c). Statistically, more than 65% of the HSC population is found within a distance of 10 μm from Caveolin-1[+] AECs (Fig. 4c; Supplementary Fig. 5d), suggesting that arteries might influence the initial colonization of fetal BM by HSCs. Our analysis also shows that these initially engrafted HSCs are preferentially located in the centre of the diaphysis (Supplementary Fig. 5f–g). Similarly, c-Kit[+] HSPCs are closely aligned with Caveolin-1[+] AECs at E16.5 (Fig. 4d–f; Supplementary Fig. 5e), suggesting a role of arteries in the colonization of fetal BM by HSCs and HSPCs. Analysis of our scRNA-seq data indicates that embryonic AECs express transcripts for niche factors[12,26,61–63], such as *Cxcl12*, *Kitl*, *Jag1* and *Jag2*, *Dll4*, *Vegfc*, and *Tgfb2*, which are known to act on multipotent HSPCs (Fig. 4g; Supplementary Fig. 6a, b). Interactome analysis further supports signaling interactions between fetal AECs and HSPCs during BM development (Fig. 4h).

**AEC-derived Wnt regulates fetal HSC and HSPC development.** Our scRNA-seq analysis also hinted at a role of Wnt signaling in the communication between ECs and HSPCs in fetal bone, but transcripts for many Wnt pathway components were barely detectable in the whole transcriptome dataset. We therefore performed targeted scRNA-seq analysis using a custom panel with pre-designed primers for Wnt pathway components (see "Methods"). This analysis revealed that AECs are a source of Wnt ligands in the E18.5 BM with Wnt2, an activator of β-catenin dependent canonical Wnt signaling[64,65], showing the highest expression (Fig. 5a). Known receptors for Wnt2, namely Frizzled-5 and Frizzled-9, are expressed by haematopoietic cells (Supplementary Fig. 7a, b)[66,67]. Interactome analysis supports potential regulation of haematopoietic cells by AEC-derived Wnt2 at E18.5 (Fig. 5b).

Wnt signaling is an evolutionary conserved pathway implicated in the control of numerous types of tissue stem cells[68,69]. To investigate a potential role of AEC-derived Wnt in HSC and HSPC development, we generated tamoxifen-controlled *Bmx-CreERT2 Wls*[lox/lox](*Wls*[iΔBmx]) knockout mice (Supplementary Table S1) to eliminate Wntless (encoded by the gene *Wls*), a protein indispensable for secretion of all Wnt family ligands, in AECs (Fig. 5c; Supplementary Fig. 7c–f). In *Wls*[iΔBmx] embryos at E18.5, the area containing c-Kit[+] haematopoietic cells is significantly reduced without alteration in bone length, which argues against unspecific developmental delay (Fig. 5d; Supplementary Fig. 7g). To test whether HSCs are reduced in *Wls*[iΔBmx] mice, we performed CD45.2/CD45.1 competitive repopulation analysis. All cells from 2 E18.5 *Wls*[iΔBmx] or littermate control femurs were used as CD45.2 donors together with $5 \times 10^5$ CD45.1 BM cells from adult CD45.1 donors for transplantation into lethally-irradiated adult CD45.1 recipients (Fig. 5e). Four months after transplantation, CD45.1 recipients transplanted with *Wls*[iΔBmx] cells show a significantly lower percentage of CD45.2[+] cells in the myeloid, B cell and T cell lineage relative to control (Fig. 5f), indicating significant reduction of HSCs in mutant embryos. To confirm the immunostaining and repopulation results, isolated E18.5 *Wls*[iΔBmx] and littermate control femurs were analysed by flow cytometry. Consistently, the number of LSK (Lin[−] Sca1[+] cKit[+]) cells/HSPCs and HSCs (CD150[+] CD48[−] LSK) are reduced in *Wls*[iΔBmx] femurs (Fig. 5g and Supplementary Fig. 7h, i). The distance of CD150[+] CD48[−] CD41[−] Lin[−] HSCs to Caveolin-1[+] arteries is significantly increased in *Wls*[iΔBmx] femurs relative to littermate control at E18.5 (Fig. 5h and Supplementary Fig. 7j, k). Similarly, the number of other Lin[−] c-Kit[+] HSPC subsets is reduced in *Wls*[iΔBmx] femurs (Fig. 5i; Supplementary Fig. 8a). In E18.5 *Wls*[iΔBmx] femur the percentage of Ter119[+] erythrocytes is reduced, whereas the percentage of myeloid and B-lymphocyte is increased, suggesting that the balance of haematopoietic cell generation in BM is disturbed (Fig. 5j). In contrast, similar changes are absent in *Wls*[iΔBmx] peripheral blood, which also receives blood cells generated in the fetal liver (Supplementary Fig. 8b). Consistent with the latter, we do not detect significant reductions of c-Kit[+] cells, the c-Kit[+] covered area or the number of other HSPCs in fetal liver (Supplementary Fig. 8c, d). When cells from embryonic *Wls*[iΔBmx] liver are analysed by competitive repopulation assay, CD45.2[+] cells in the myeloid, B cell and T cell lineage are not significantly different relative to littermate control (Supplementary Fig. 8e, f). Likewise, when

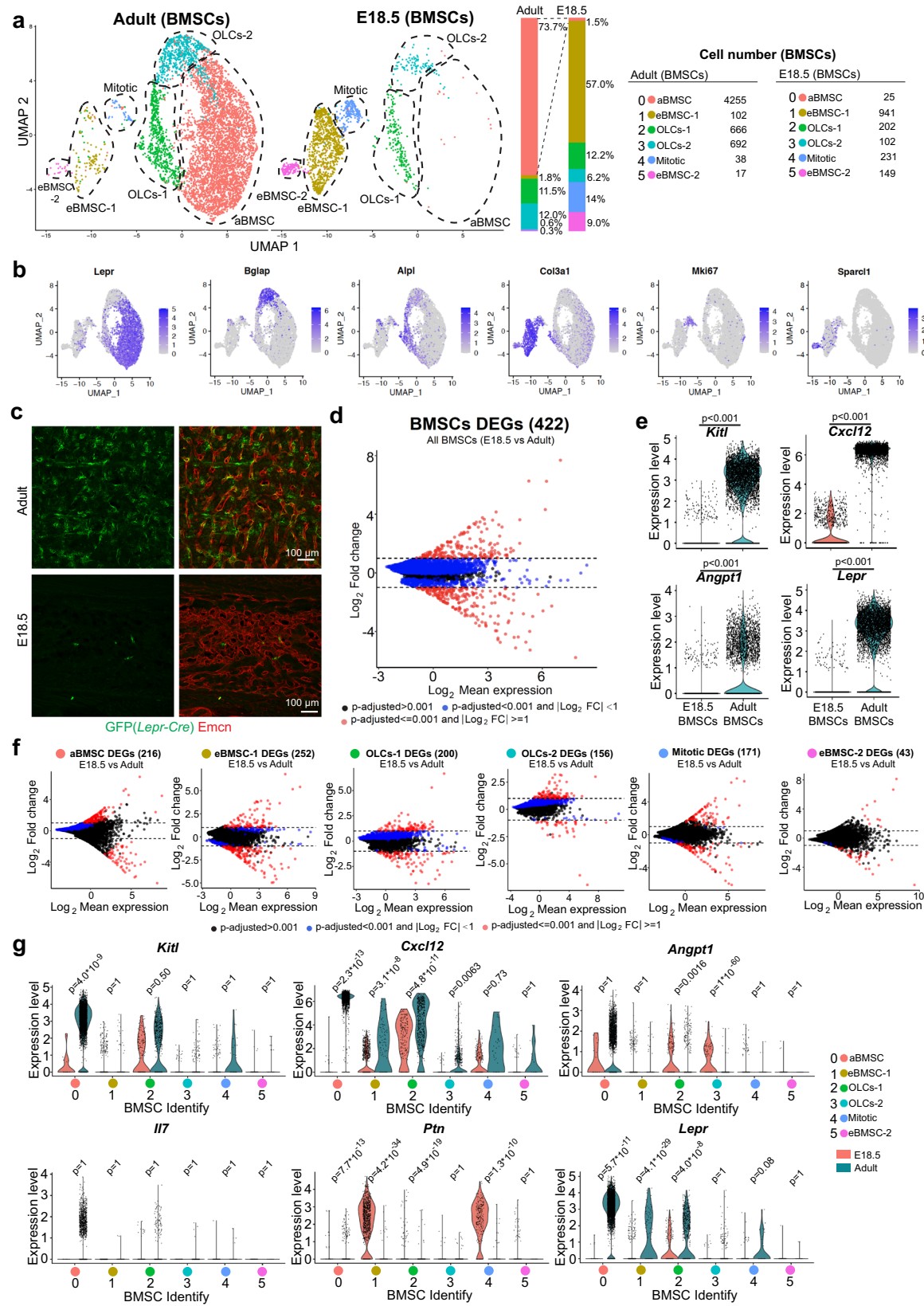

cells from $Wls^{i\Delta Bmx}$ fetal liver are transplanted into lethally-irradiated mice, haematopoietic reconstitution efficiency is not significantly altered (Supplementary Fig. 8g, h). These results argue that alterations in $Wls^{i\Delta Bmx}$ femoral BM are not caused by defective HSPC expansion in liver. In contrast to the defects seen in AEC-specific $Wls^{i\Delta Bmx}$ mutants, tamoxifen-controlled blockade of Wnt ligand release from vascular smooth muscle cells (vSMCs), which cover larger arteries, in $Myh11$-$CreER$ $Wls^{lox/lox}$ ($Wls^{i\Delta Myh11}$) mutants do not affect HSCs and c-Kit[+] haematopoietic cells (Supplementary Fig. 9a, b). Thus, AECs but not vSMCs are a relevant source of Wnt ligands during fetal BM development.

**Fig. 2 Comparative analysis of embryonic and adult bone marrow stromal cells. a** UMAP plot showing unsupervised subclustering of E18.5 and adult bone marrow stromal cells (BMSCs). Bar chart indicates percentage and panel on the right shows the number of cells in each subcluster. **b** UMAP plot showing distribution of representative markers for each BMSC subcluster. Color represents relative expression level of the gene. **c** Representative projected confocal images showing the number of *Lepr-Cre*-labeled perivascular cells in adult or E18.5 *R26-mTmG* Cre reporter bone marrow. *Lepr-Cre*-controlled GFP expression predominantly labels sinusoidal vessel-associated (reticular) cells in adult BM, whereas GFP+ cells are very sparse at E18.5. ECs, Emcn (red). **d** Differentially expressed genes (DEGs) between all BMSCs cells at E18.5 and in adult. Blue dots, *p*-adjusted value < 0.001; Red dots, *p*-adjusted value < 0.001 and Log$_2$ fold change >1. Two-tailed Student's *t* test. **e** Violin plots showing expression of critical niche factors in all BMSCs at E18.5 or in adult. The *y* axis indicates expression level of each gene with log normalized value (see "Methods"). Two-tailed Student's *t* test. **f** DEGs for each E18.5 and adult BMSC subcluster. Blue dots, *p*-adjusted value < 0.001; Red dots, *p*-adjusted value < 0.001 and Log$_2$ fold change >1. Two-tailed Student's *t* test. **g** Violin plots showing expression of HSPC niche-related genes in each E18.5 or adult BMSC subcluster.

**BM innervation is not critical for fetal HSPC development**. Next, we investigated how AEC-derived Wnt ligands might influence HSPC expansion. We found that an artery located near the lesser trochanter of femur (defined as trochanter artery) extends into BM from E15.5 onward (Supplementary Fig. 10a–d). Tuj1+ (neuron-specific class 3 beta-tubulin) nerve fibres are closely aligned with the extending trochanter artery and invade the femoral BM after E16.5 (Supplementary Fig. 10e–g). Notably, the distal front of Tuj1+ nerve fibres trail the extending artery (Supplementary Fig. 10f, g), suggesting that AECs might guide nerve growth. Indeed, *Wls*$^{i\Delta Bmx}$-induced blockade of Wnt secretion from AECs also results in delayed nerve extension into the developing femur (Supplementary Fig. 10h), a defect that is not seen in *Wls*$^{i\Delta Myh11}$ mutants (Supplementary Fig. 9c). In the adult, sympathetic nerves control HSC behavior by providing adrenergic signals acting on stromal niche cells and dopaminergic signals acting on HSPCs[12,14,17]. To investigate whether the loss of HSPC in *Wls*$^{i\Delta Bmx}$ mutants is secondary to delayed nerve ingrowth, we inactivated β-catenin (encoded by *Ctnnb1*), a critical transcriptional regulator mediating canonical Wnt signaling, in Wnt1+ cells. In *Wnt1-Cre R26-mTmG* (*Wnt1-mTmG*) mice (Supplementary Table S1), GFP expression labels Wnt1+ cells and their descendants. The resulting GFP expression labels nerves in embryonic bone, as indicated by co-localization of GFP with several neuronal markers. These GFP+ nerves enter the femur at E17.5 and extend significantly into the developing BM in neonates (Supplementary Fig. 10i–n). In E18.5 *Wnt1-Cre*$^{+/T}$ *Ctnnb1*$^{lox/lox}$ (*Ctnnb1*$^{\Delta Wnt1}$) mutants (Supplementary Table S1), the nerve is significantly shorter relative to littermate control mice, whereas the area covered by c-Kit+ cells is not altered (Supplementary Fig. 10o, p). This result indicates that delayed nerve extension is not the underlying cause of HSPC defects in *Wls*$^{i\Delta Bmx}$ mutants.

**AECs regulate HSPC proliferation through β-catenin**. Next, we addressed whether AEC-derived Wnt signals can influence HSPC function directly. Flow cytometric analysis of E18.5 embryos carrying a *Tcf/Lef:H2B*-GFP reporter allele[70] indicates GFP expression in LSK cells (Supplementary Fig. 11a). Likewise, scRNA-seq analysis reveals the expression of several Wnt pathway components in HSPCs (Supplementary Fig. 11b). For functional studies, we generated *Vav1-Cre*$^{+/T}$ *Ctnnb1*$^{lox/lox}$ (*Ctnnb1*$^{\Delta Vav1}$) mutants lacking β-catenin function in Vav1+ haematopoietic cells. At E18.5, the c-Kit+ cell area as well as the number of HSCs, LSK cells and other Lin− c-Kit+ HSPC are reduced in *Ctnnb1*$^{\Delta Vav1}$ femur relative to littermate control (Figs. 6a–d). To circumvent potential defects in early haematopoietic development and confirm the necessity of β-catenin-dependent Wnt signaling in c-Kit+ HSPCs, we next generated tamoxifen-inducible *Kit-CreER*$^{+/T}$ *Ctnnb1*$^{lox/lox}$ (*Ctnnb1*$^{i\Delta Kit}$) mice. Following tamoxifen administration from E14.5-E17.5, *Ctnnb1*$^{i\Delta Kit}$ mutants show reduction of the c-Kit+ area in femur, reduced number of HSCs, LSK cells and other HSPCs (Fig. 6e–h;

Supplementary Fig. 11c). Importantly, *Kit-CreER* induces only limited recombination in Emcn+ ECs in embryonic femur (Supplementary Fig. 11d)[71], further arguing for a role of β-catenin in HSPCs. Indicating that the reduction of HSPCs is not caused by the knock-in of CreER into the *c-Kit* locus itself, the number of HSPCs is not significantly altered in *Ctnnb1*$^{i\Delta Kit}$ embryos without tamoxifen treatment (Supplementary Fig. 11e). The sum of these results indicates that AEC-controlled Wnt/β-catenin signaling promotes HSPC expansion in fetal BM.

Arguing further for the regulation of HSPCs by Wnt ligands, we detect significantly reduced EdU incorporation into *Wls*$^{i\Delta Bmx}$ c-Kit+ haematopoietic cells and LSK cells from E18.5 femur (Fig. 6i, j). Similar EdU incorporation defects are seen in *Ctnnb1*$^{\Delta Vav1}$ and *Ctnnb1*$^{i\Delta Kit}$ mutant LSK cells relative to the corresponding littermate control samples (Fig. 6k, l). Treatment of FACS-isolated neonatal Lin− c-Kit+ cells with the Wnt/β-catenin signaling inhibitor endo-IWR-1 (IWR-1) significantly inhibits colony formation in vitro (Supplementary Fig. 12a). IWR-1-treated Lin− cells are more abundantly found in G$_0$ phase and reduced at the G$_1$/S and G$_2$/M checkpoints (Supplementary Fig. 12b). Wnt inhibition also reduces the intensity and distribution of nuclear Cyclin D1 immunostaining in neonatal Lin− c-Kit+ cells in vitro (Supplementary Fig. 12c).

Ex vivo treatment of HSPCs with Wnt2, the major Wnt ligand expressed by AECs in fetal BM (Fig. 5a), promotes EdU incorporation and colony formation ability of FACS-isolated LSK cells (Fig. 7a, b). Furthermore, repeated Wnt2 stimulation of cultured adult or neonatal LSK cells significantly increases the absolute number of HSCs and LSK cells as well as total cells (Fig. 7c, d; Supplementary Fig. 13a, b). Next, we added Wnt2 to cultured GFP+ LSK cells sorted from *Vav1-Cre Rosa26-mTmG* double transgenic mice, in which descendant haematopoietic cells are genetically (and irreversibly) labeled by expression of membrane-anchored GFP, prior to transplantation into lethally-irradiated WT mice (Fig. 7e). In recipient animals, the percentage of GFP+ cells and GFP+ myeloid cells are significantly increased after Wnt2 pre-treatment (Fig. 7f–h; Supplementary Fig. 13c). Similarly, haematopoietic reconstitution at 2 weeks after transplantation is significantly increased when donor cells were pre-treated with Wnt2 (Fig. 7i, j; Supplementary Fig. 13d). Together, the results above indicate that AEC-controlled Wnt/β-catenin signaling promotes colonization of bone by HSPCs in the embryo. Furthermore, we show that Wnt2, a ligand that is highly expressed by AECs in fetal BM, increases HSPC proliferation and colony formation.

**Discussion**

The sum of our work provides insight into a little understood biological process, namely the de novo colonization of fetal bone by HSPCs[9], which is essential for the development of a fully functional haematopoietic system. Our single cell transcriptomic analysis reveals substantial differences between fetal and adult BM with regard to cellular composition, gene expression and cell-

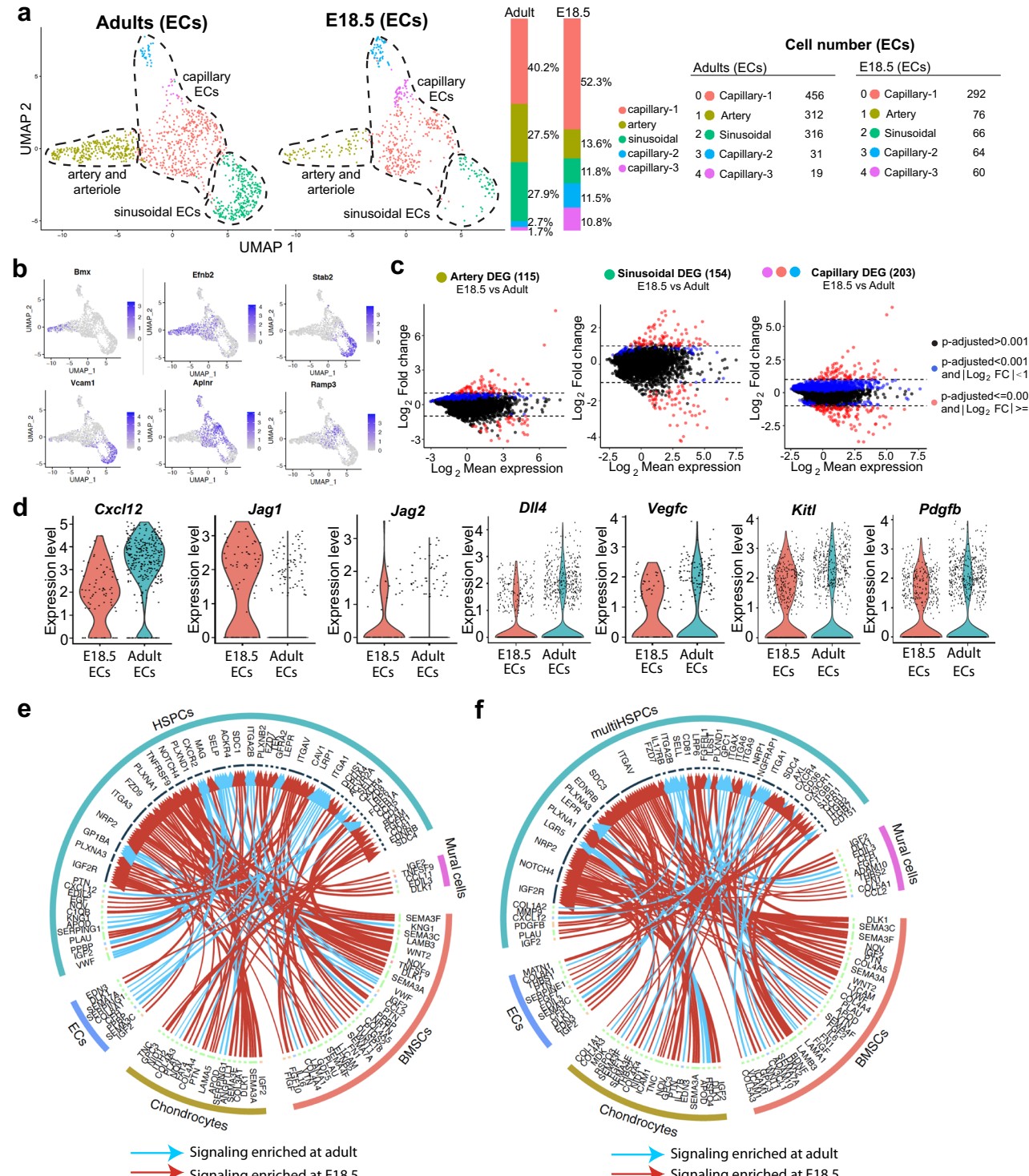

**Fig. 3 Comparative analysis of embryonic and adult endothelial cells. a** UMAP plot showing unsupervised subclustering of E18.5 and adult femoral ECs. Bar charts indicate percentage of each subcluster. Cell numbers are shown on the right. **b** UMAP plots showing distribution of representative markers for each EC subcluster. Color represents relative gene expression level. **c** DEGs for E18.5 and adult EC subclusters. Blue dots, *p*-adjusted value < 0.001; Red dots, *p*-adjusted value < 0.001 and $Log_2$ fold change >1. Two-tailed Student's *t* test. **d** Violin plots showing expression of selected EC-derived HSPC-supporting factors at E18.5 or in adult with log normalized values (*y* axis). **e**, **f** Interactome analysis of potential interactions (arrows) between non-haematopoietic supportive cells and HSPCs (**e**) or multipotent HSPCs (**f**). Width of line and arrow represent relative log2 fold change of genes. Cyan arrows indicate signaling enriched in adult; red arrows show enrichment at E18.5.

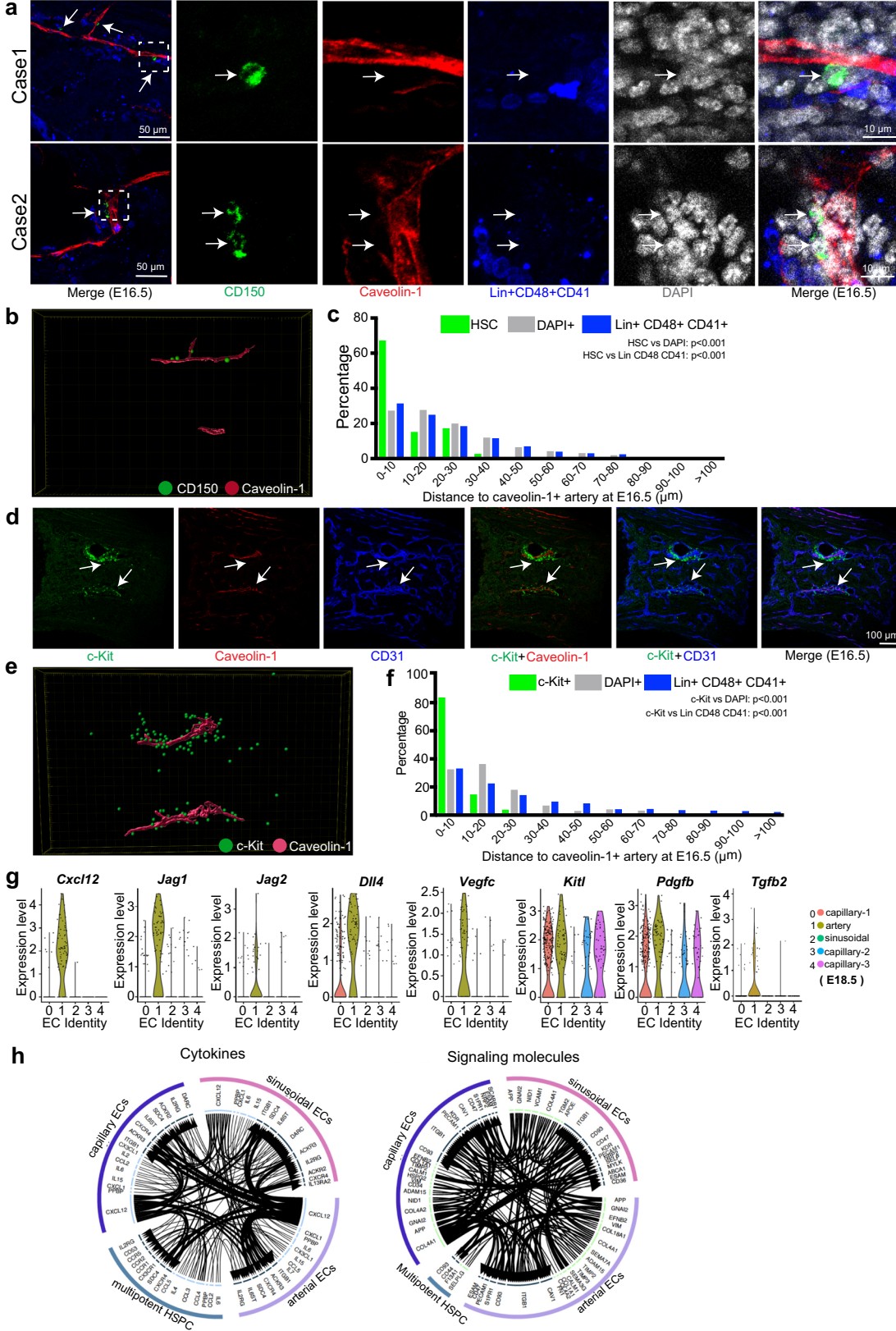

cell interactions. LepR$^+$ reticular cells, a stromal cell population that is associated with sinusoidal vessels and provides essential niche signals for HSPCs in the adult organism[18,19,55], are absent in fetal marrow. This finding is consistent with previous publications using immunostaining and FACS to investigate BMSC development[54–56], but our scRNA-seq analysis suggests that

absence of the LepR$^+$ population reflects not just absent expression of a single gene, namely *Lepr*, but the actual lack of a BMSC subcluster in fetal BM. As LepR$^+$ perivascular cells become abundant at 2 weeks after birth, one possibility is that their function is taken over by another population of perivascular mesenchymal cells in the embryo and during early postnatal

**Fig. 4 Arteries are linked to initial HSC and HSPC engraftment in fetal BM. a** Overview and high-magnification representative images showing close alignment of Lin⁻ CD48⁻ CD41⁻ CD150⁺ HSC (green, arrows) and Caveolin-1⁺ AECs (red) at E16.5. Blue, Lin⁺ CD48⁺ CD41⁺. White box in left panel indicates position of high magnification images. **b** 3D-reconstructed image showing CD150⁺ cells (green) and Caveolin-1⁺ arteries (red). **c** 3D-reconstruction-based quantification of distance between HSCs ($N = 48$), Lin⁺ CD48⁺ CD41⁺ mature haematopoietic cells ($N = 2203$) or DAPI⁺ random cells ($N = 6174$) and Caveolin-1⁺ AECs at E16.5. P-value, Kolmogorov-Smirnov test. **d** Representative images showing close alignment of c-Kit⁺ haematopoietic cells (green) and Caveolin-1⁺ AECs (arrows, red) at E16.5. CD31, blue. **e** Representative 3D-reconstructed image showing location of c-Kit⁺ haematopoietic cells (green) and Caveolin-1 (red). **f** 3D-reconstruction-based quantification of distance between c-Kit⁺ ($N = 258$), Lin⁺ CD48⁺ CD41⁺ mature haematopoietic cells ($N = 2052$) or DAPI⁺ random cells ($N = 6665$) and Caveolin-1⁺ AECs at E16.5. P-value, Kolmogorov-Smirnov test. **g** Violin plots showing expression of selected genes encoding angiocrine or HSPC-supportive factors in AECs or other EC subclusters at E18.5 with log normalized values (y axis). **h** Interactome analysis of potential interactions (arrows) between multipotent HSPCs and different ECs subclusters. Width of line and arrow represent relative expression level of genes.

development. Our work identifies eBMSC-1 as the most abundant perivascular cell population in fetal BM, but these cells do not extend overt reticular fibres and lack the expression of major niche factors, such as SCF, Cxcl12 or angiopoietin-1. A small population of eBMSC-1 cells is also found in adult BM, but these cells only represent a small fraction of total BMSCs and there are substantial differences in gene expression between eBMSC-1 cells found at E18.5 and their adult counterpart. While future work will have to address the function and differentiation potential of the eBMSC-1 population, it is possible that these cells are immature and might give rise to other BMSC populations. It is also notable that, with the exception of LepR⁺ reticular cells, other major cellular components of adult BM stroma are already present in the fetus and might therefore support haematopoietic colonization of marrow.

Numerous studies have investigated the niche microenvironment of HSCs in adult bone. It has been proposed that arteriolar niches maintain HSC quiescence in the adult[61,72], whereas several other studies place quiescent, long-term adult HSCs in the proximity of sinusoidal vessels and thereby LepR⁺ reticular niche cells[73,74]. While our findings do not rule out potential contributions of sinusoidal vessels, osteoblastic cells and other stromal cell populations, we show that LepR⁺ reticular cells are absent in fetal BM and that embryonic HSCs and HSPCs are associated with the artery in the developing femur. Arteries are involved in the development of the haematopoietic system at multiple levels. In the early embryo, the aorta is a major site for emergence of definitive HSCs through endothelial-haematopoietic transition and, later in development, arterioles help to maintain HSCs in the fetal liver[4,6,8]. Our findings add further light on the role of arteries by showing that they promote the first haematopoietic colonization of fetal BM by activating canonical, β-catenin-dependent Wnt signaling in HSPCs. AEC-derived Wnt enhances HSPC expansion and proliferation but, as we see a reduction of these cells in E16.5 femur, additional roles in the circulation, entry or engraftment of fetal HSPCs should not be ruled out. In addition to AECs, other stromal cell populations might contribute to the haematopoietic colonization of fetal BM, which may involve the release of Wnt ligands but also other molecular signals. This might potentially explain why the phenotypes seen in Ctnnb1^ΔVav1 or Ctnnb1^iΔKit mutants are more severe than the haematopoietic defects in Wls^iΔBmx embryos. Canonical and non-canonical Wnt signaling has been previously implicated in other aspects of HSPC function[69,75–77], which is consistent with our findings.

Our work closes an important knowledge gap in the development of the haematopoietic system between haematopoiesis in the fetal liver[8] and the establishment of the mature marrow in postnatal life. Our findings raise multiple interesting questions that deserve further investigation. What are the cellular and molecular changes mediating the relocation of HSCs from arteries towards sinusoidal vessels in postnatal marrow? What is the exact nature of eBMSCs and do these cells give rise to other mesenchymal cell populations found in adult BM, and could this process be repeated after irradiation and BM transplantation? As irradiation-based conditioning for HSPC transplantation leads to loss of LepR⁺ cells[78], it will be also interesting to evaluate whether AEC-derived Wnt and other signals might drive transplant engraftment and expansion, as it has been shown for SCF[61]. Thus, findings in fetal marrow may potentially open up new conceptual avenues for BM regeneration.

## Methods

**Animal experiments and genetically modified mice**. All experiments were performed according to the institutional guidelines and laws, following the protocols approved by animal ethics committees and covered by permissions granted by the Landesamt fuer Natur, Umwelt und Verbraucherschutz (LANUV) of North Rhine-Westphalia, Germany.

Following overnight mating, female mice were examined in the following morning for the presence of a vaginal plug, which was counted as embryonic day 0.5 (E0.5). C57BL/6 mice were used for all analysis of wild-type BM. The developmental stage of embryos was confirmed by morphological features. Mice were typically sacrificed between 8 am and 10 am local time. Littermates with appropriate genotypes were used as controls for mutants whenever possible.

The transgenic mouse models used in this study are summarized in Supplementary Table 1. Efnb2^GFP[79], TCF/Lef:H2B-GFP[70], Cdh5-membrane-tdTomato H2B-EGFP (Cdh5-mTnG)[80] reporters were used for the visualization of AECs, Wnt signaling and EC surface and nuclei, respectively. In Rosa26-mTmG[81] reporter mice, Cre activity leads to an irreversible switch from constitutive expression of membrane-anchored tdTomato protein to membrane-anchored GFP. Rosa26-mTmG Cre reporter animals were interbred with Bmx-CreERT2[82] and Wnt1-Cre[83] mice for genetic lineage tracing or the labeling of cells. For genetic lineage tracing, 1 mg tamoxifen (Sigma, T5648) was administered to pregnant dams at E14.5(Bmx-mTmG). Mice were analysed at the time point indicated in the figures and legends.

Floxed Wls(Wls^tm1.1Lan)[84] conditional mutants were interbred with Myh11-cre/ERT2^ISoff/J[85] or Bmx-CreERT2 mice to generate Wls^iΔMyh11 (Myh11-CreER^+/T Wls^f/f; littermate control Myh11-CreER^+/+ Wls^f/f) and Wls^iΔBmx(Bmx-CreER^+/T Wls^f/f; littermate control Bmx-CreER^+/+ Wls^f/f) animals, respectively. For smooth muscle cell-specific gene inactivation (Wls^iΔMyh11), 3 mg tamoxifen were injected intraperitoneally to pregnant dams once a day from E14.5 to E16.5. For AEC-specific gene inactivation (Wls^iΔBmx), pregnant dams received 2 mg tamoxifen intraperitoneally once a day from E14.5 to E16.5.

Floxed Ctnnb1 (Ctnnb1^tm2Kem)[86] conditional mutants were interbred with Wnt1-Cre, Vav1-Cre[87] or Kit-CreER[71] mice to generate Ctnnb1^ΔWnt1 (Wnt1-Cre^+/T Ctnnb1^f/f; littermate control Wnt1-Cre^+/+ Ctnnb1^f/f), Ctnnb1^ΔVav1 (Vav1-Cre^+/T Ctnnb1^f/f; littermate control Vav1-Cre^+/+ Ctnnb1^f/f) or Ctnnb1^iΔKit (Kit-CreER^+/T Ctnnb1^f/f; littermate control Kit-CreER^+/+ Ctnnb1^f/f) mutants. For Ctnnb1^iΔKit mice, 2 mg tamoxifen were injected intraperitoneally to pregnant dams once a day from E14.5 to E17.5.

All animals were maintained in a predominantly C57BL/6 background either by interbreeding (for conditional mutants) or by backcrossing to C57BL/6 wild-type mice (for Cre/CreERT2 lines and reporter alleles but also for the expansion of newly imported or freshly generated lines). Mice were routinely genotyped by PCR. Protocols and primer sequences can be provided upon request. All the animals were housed in the animal facility at the Max Planck Institute for Molecular Biomedicine.

**Flow cytometry**. Information about primary antibodies is summarized in Supplementary Table 2.

Embryonic femurs were dissected under a microscope and crushed by pestle repeatedly before cells were collected in 2% FCS-PBS solution. Tissues were immersed

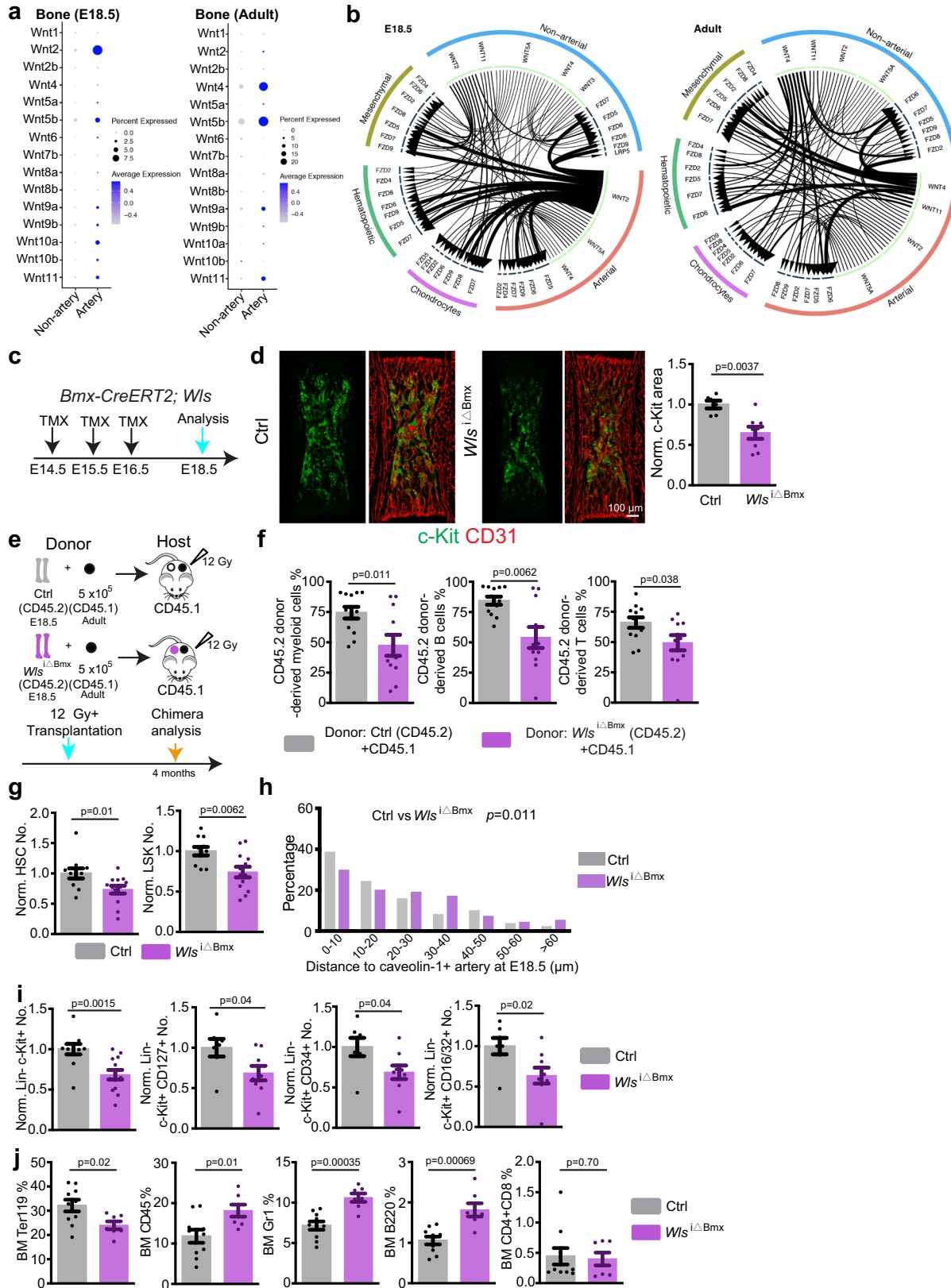

in 2 ml dissociation solution (2% FCS-PBS solution with approximate 145 U/ml type 4 Gibco collagenase) and incubated at 37 °C for 30 min. Samples were filtered using 50 μm Cell Trics (Sysmex, 04-0042-2317) to obtain single cell suspensions. Before and after the passage of bone cells, 1 ml FCS-PBS was used to wash the Cell Trics. Primary antibodies were diluted in 2% FCS-PBS solution and incubated with cells on ice for 30 min. Cells were washed 1 time by 2% FCS-PBS solution and incubated with

secondary antibodies for 30 min. Cells were resuspended in 2% FCS-PBS supplemented with 1 μg/ml DAPI to allow exclusion of nonviable cells when required and used for flow cytometry. Adult bones were treated with similar conditions for FACS staining. Cell analysis was performed using a BD FACSVerse.

Livers samples were gently crushed by pestle before cells were dissociated, filtered and stained using the same procedure as for bone marrow cells.

**Fig. 5 AEC-HSPC crosstalk through Wnt in fetal BM. a** Targeted scRNA-seq expression analysis showing different Wnt ligands in E18.5 and adult non-arterial or arterial ECs. Dot size indicates percentage of cells where the gene was detected, color represents the average expression level of the gene in each cluster. **b** Interactome analysis depicting potential Wnt signaling interactions in E18.5 and adult BM. Width of line and arrow represent relative expression level of genes. **c** Diagram depicting tamoxifen treatment and analysis of $Wls^{i\Delta Bmx}$ mice. **d** Representative overview image of c-Kit$^+$ cells in E18.5 $Wls^{i\Delta Bmx}$ mutant and littermate control. Graph shows quantification of c-Kit$^+$ cell covered area. Ctrl = 6; $Wls^{i\Delta Bmx}$ = 8. Error bars, mean ± s.e.m. P values, two-tailed unpaired Student's t Test. **e** Schematic protocol for CD45.2/CD45.1 long term-repopulating assay. All cells from two $Wls^{i\Delta Bmx}$ or littermate control femurs were used as CD45.2 donor. **f** Quantification of long-term competitive repopulation assay showing percentage of donor-derived (CD45.2, control = 12 and $Wls^{i\Delta Bmx}$ = 11) myeloid cells (CD11b$^+$), B cells (B220$^+$), and T cells (CD4$^+$ and CD8$^+$). Error bars, mean ± s.e.m. Two-tailed Student's t test. **g** FACS-based quantification of normalized HSC and LSK cell numbers (Ctrl = 11; $Wls^{i\Delta Bmx}$ = 13) in $Wls^{i\Delta Bmx}$ and littermate control BM at E18.5. Absolute cell numbers were normalized to control. Error bars, mean ± s.e.m. Two-tailed Student's t test. **h** 3D-reconstruction-based quantification of distance between HSCs and Caveolin-1$^+$ AECs at E18.5 in $Wls^{i\Delta Bmx}$ (N = 102) and littermate control mice (N = 272). P-value, Kolmogorov-Smirnov test. **i** FACS-based quantification of Lin$^-$ c-Kit$^+$ (Ctrl = 11; $Wls^{i\Delta Bmx}$ = 13) and of Lin$^-$ c-Kit$^+$ CD127$^+$, Lin$^-$ c-Kit$^+$ CD34$^+$, Lin$^-$ c-Kit$^+$ CD16/32$^+$ cell number (Ctrl = 7; $Wls^{i\Delta Bmx}$ = 9) in $Wls^{i\Delta Bmx}$ and littermate control BM at E18.5. Absolute cell numbers were normalized to control. Error bars, mean±s.e.m. Two-tailed Student's t test. **j** FACS-based quantification of Ter119$^+$, CD45$^+$, Gr-1$^+$, B220$^+$, CD4/8$^+$ cell percentage in femoral BM of $Wls^{i\Delta Bmx}$ and littermate control at E18.5. Error bars, mean ± s.e.m. Two-tailed Student's t test.

**Single cell RNA-seq library preparation and sequencing**. Mouse fetal femur single cells were obtained as described above in the flow cytometry section and stained with biotin-conjugated antibodies for Lin (Ter119, Gr-1, B220, CD3e) (BD, 559971) followed by negative selection using streptavidin-coated magnet beads (MiltenyiBiotec, 130-105-637). To deplete lineage committed hematopoietic cells in adult BM, whole BM cells were incubated with lineage cocktail antibody (Miltenyi Biotec, 130-090-858, 10 µl per $10^7$ cells) and separated using LS columns (Miltenyi Biotec, 130-042-401) via MACS Separator. The remaining Lin- cells were further incubated with biotinylated anti-CD71 antibody (Biolegend, 113803, 10 µl per $10^7$ cells) and Anti-biotin Microbeads (Miltenyi Biotec, 130-105-637, 20 µl of per $10^7$ cells) before separation with an LS column. The remaining cells were then incubated with CD117 (Miltenyi Biotec, 130-091-224, 10 µl per $10^7$ cells), CD45 (Miltenyi Biotec, 130-052-301, 10 µl per $10^7$ cells) and Ter119 (Miltenyi Biotec, 130-049-901, 10 µl for $10^7$ cells) antibodies for a final lineage separation to maximize the cells of interest. Single cells were counted using Luna-II automated cell counter (Logos Biosystems) and E18.5 or adult were loaded on a microwell cartridge of the BD Rhapsody Express system (BD) following the manufacturer's instruction. Single cell whole transcriptome analysis libraries were prepared according to the manufacturer's instructions using BD Rhapsody WTA Reagent kit (BD, 633802) and sequenced on the Illumina NextSeq 500 using High Output Kit v2.5 (150 cycles, Illumina) for 2 × 75 bp paired-end reads with 8 bp single index aiming sequencing depth of >20,000 reads per cell for each sample.

**Single cell RNA-seq data analysis**. Sequencing data were processed with UMI-tools (version 1.0.1), aligned to the mouse reference genome (mm10) with STAR (version 2.7.1a), and quantified with Subread featureCounts (version 1.6.4). Alternatively, we also processed the FASTQ format of sequencing raw data with BD Rhapsody WTA Analysis pipeline (version 1.0) on SevenBridges Genomics online platform (SevenBridges) especially for the quality control. After the sequencing data quality control and genome alignment, we recovered 16,171 reads per cell for adult BMSCs and 10,203 reads per cell for E18.5 BMSCs. The sequencing saturation percentages were 87.69% and 80.06%, respectively.

Data normalization, detailed analysis and visualization were performed using Seurat (version 3.1.3) if not specified otherwise. For initial quality control of the extracted gene-cell matrices, we filtered cells with parameters nFeature_RNA > 200 & nFeature_RNA < 5000 for number of genes per cell and percent.mito < 10 for percentage of mitochondrial genes and genes with parameter min.cell = 3. Filtered matrices were normalized by LogNormalize method with scale factor = 10,000. Variable genes were found with parameters of mean.function = ExpMean, dispersion.function = LogVMR, x.low.cutoff = 0.0125, x.high.cutoff = 3 and y.cutoff = 0.5, trimmed for the genes related to cell cycle (GO:0007049) and then used for principal component analysis. FindIntegrationAnchors and IntegrateData with default options were used for the data integration. Statistically significant principal components were determined by JackStraw method and the first 11 principal components were used for UMAP non-linear dimensional reduction.

Unsupervised hierarchical clustering analysis was performed using FindClusters function in Seurat package. We tested different resolutions between 0.1~0.9 and selected the final resolution using clustree R package to decide the most stable as well as the most relevant for our previous knowledges. Cellular identity of each cluster was determined by finding cluster-specific marker genes using FindAllMarkers function with minimum fraction of cells expressing the gene over 25% (min.pct = 0.25), comparing those markers to known cell type-specific genes from previous studies and further confirmed using the R package SingleR, which compares the transcriptome of each single cell to reference datasets to determine cellular identity.

For subclustering analysis, we isolated specific cluster(s) using subset function, extracted data matrix from the Seurat objectusing GetAssayData function and repeated the whole analysis pipeline from data normalization.

Differentially expressed genes (DEGs) were identified using the non-parametric Wilcoxon rank sum test by FindMarkers function of Seurat package. We used default options for the analysis if not specified otherwise. Results were visualized using EnhancedVolcanoR package (version 1.10.0). FeaturePlot, VlnPlot and DotPlot functions of Seurat package were used for visualization of selected genes. The "VlnPlot" function of Seurat package was used for violin plots to show the expression level of selected genes with log normalized value by default.

The MetaCellR package was used to compute cell-to-cell similarity, to compute k-NN graph covers and derive distribution of RNA in small homogeneous groups of cells, which are defined as metacells and to derive strongly separated clusters using bootstrap analysis and computation of graph covers on resampled data. We removed mitochondrial genes (annotated with the prefix '^mt-') and cells in which <800 UMIs were detected from the matrix. Gene features were selected using the threshold value for normalized var/mean (T_vm) = 0.08 and a minimum total UMI count >100. We computed a similarity graph using balanced K-nn graph with K = 100, resampled to 500 metacell partitions (n_resamp = 500), each covering 75% of the cells (p_resamp = 0.75), performed bootstrap to generate co-clustering graph with the number of associated neighbors > 30 (K = 30) and minimum metacell size > 30 (min_mc_size = 30) and removed outlier cells that strongly deviate from their metacell's expression profile using mcell_mc_split_filt function. Previously identified top marker genes for each cluster in Seurat objects were used for clustering and coloring of metacells. The results were visualized using mcell_mc2d_plot, the two-demensional projection function of MetaCell (version 0.3.6).

An R package iTALK (https://doi.org/10.1101/507871) (version 0.1.0) was used for the ligand-receptor interactome analysis. For the analysis of each single sample, top 50% of genes in their mean expression values were selected and used for ligand-receptor pair identification using FindLR function with datatype = mean count. For comparative interactome analysis, differentially expressed genes between samples were identified in each cell type using the non-parametric Wilcoxon rank sum test and used for the detection of ligand-receptor pairs using FindLR function. The results were plotted by LRPlot function with datatype = DEG. Metadata for each cell (Supplementary Data 1), cluster-specific markers (Supplementary Data 2) and matrics summary of each experiment (Supplementary Data 3) are provided with this paper.

**Targeted single-cell RNA-seq analysis of Wnt pathway genes**. We produced a custom targeted gene panel on the BD Rhapsody platform containing a total of 153 genes including Wnt ligands and receptors together with representative cell type-specific markers (Supplementary Data 4). For E18.5, we sacrificed 8 embryos to obtain Lin$^-$ cells and another 8 embryos to isolate CD31$^+$ ECs by FACS. For adult, we sacrificed 4 mice to obtain Lin$^-$ cells. Cells were loaded on microwell cartridges of the BD Rhapsody Express system (BD) and processed separately. BD Rhapsody Targeted Amplification kit (BD, 633734) was used for library preparation. Sequencing was performed on the Illumina NextSeq 500 using Mid Output Kit v2.5 (150 cycles, Illumina) for 2 × 75 bp paired-end reads with 8 bp single index aiming sequencing depth of >5000 reads per cell for each sample. The FASTQ format of sequencing raw data was processed with the BD Rhapsody Targeted Analysis pipeline (version 1.0) on the SevenBridges Genomics online platform (Seven-Bridges). After the sequencing data quality control and genome alignment, we recovered 12,884 cells for Lin$^-$ and 4043 cells for ECs in E18.5 samples and 7948 Lin$^-$ cells for adult samples. Sequencing saturation percentages were 94.93%, 97.97% and 87.31%, respectively. Seurat package was used for the data processing and iTALK for the interactome analysis as described above. However, integration of multiple datasets was not performed because of the small number of detected genes to be used as anchors and each data was analyzed separately.

**Cryosections, immunohistochemistry, confocal imaging and 3D-reconstruction**. Information about primary antibodies is summarized in Supplementary Table 2.

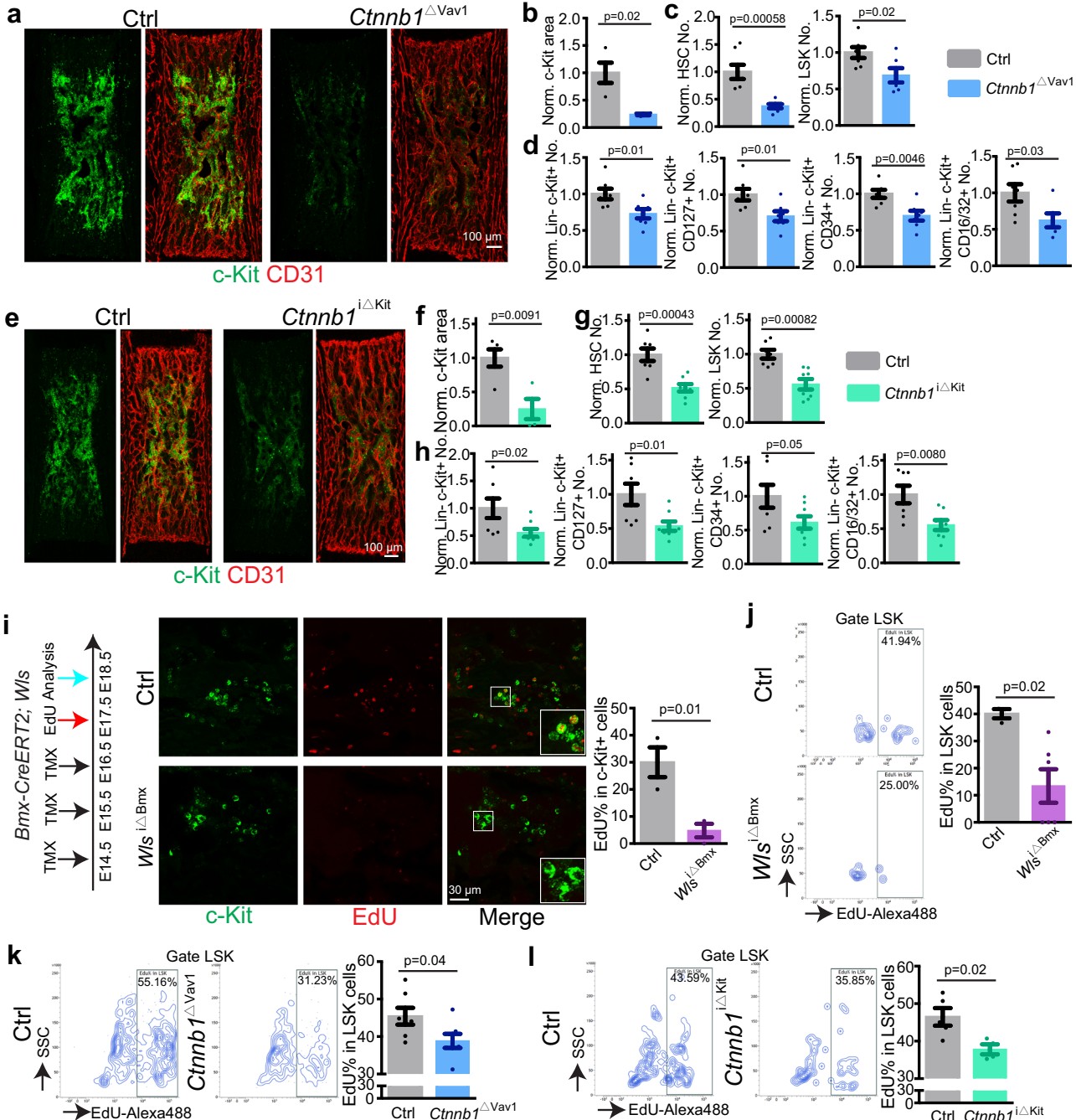

**Fig. 6 AECs regulate HSPC proliferation through Wnt and β-catenin. a** Overview images showing c-Kit+ cell covered area in *Ctnnb1*△Vav1 knockout and littermate control BM at E18.5. **b** Quantification of c-Kit+ cell covered area (Ctrl = 4; *Ctnnb1*△Vav1 = 4). Error bars, mean ± s.e.m. Two-tailed Student's *t* test. **c**, **d** FACS-based quantification of normalized HSC and LSK cells (**c**) and of Lin− c-Kit+, Lin− c-Kit+ CD127+, Lin− c-Kit+ CD34+, Lin− c-Kit+ CD16/32+ cells in *Ctnnb1*△Vav1 and littermate controls at E18.5 (**c**). Ctrl = 7; *Ctnnb1*△Vav1 = 7. Absolute cell numbers were normalized to control. Error bars, mean ± s.e.m. Two-tailed Student's *t* test. **e** Representative overview images of c-Kit+ cell covered area in *Ctnnb1*i△Kit and littermate control BM at E18.5. **f** Quantification of c-Kit+ cell covered area. Ctrl = 5; *Ctnnb1*i△Kit = 4. Error bars, mean ± s.e.m. Two-tailed Student's *t* test. **g**, **h** FACS-based quantification of normalized HSC and LSK cells (**g**) and of Lin− c-Kit+, Lin− c-Kit+ CD127+, Lin− c-Kit+ CD34+, Lin− c-Kit+ CD16/32+ cells (**h**) in *Ctnnb1*i△Kit mutants and littermate controls at E18.5. Ctrl = 7; *Ctnnb1*i△Kit = 8. Absolute cell numbers were normalized to control. Error bars, mean ± s.e.m. Two-tailed Student's *t* test. **i** Diagram showing tamoxifen injection, EdU injection and analysis of *Wls*i△Bmx mutants. High magnification single plane images showing EdU labeling of c-Kit+ cells. Image-based quantification of EdU% in c-Kit+ cells. Ctrl = 3; *Wls*i△Bmx = 3. Error bars, mean ± s.e.m. Two-tailed Student's *t* test. **j** Representative FACS plot showing EdU incorporation into *Wls*i△Bmx and control LSK cells. FACS based quantification of EdU% in *Wls*i△Bmx (*N* = 6) and control (*N* = 3) LSK cells. Error bars, mean ± s.e.m. Two-tailed Student's *t* test. **k** FACS plot showing EdU incorporation into *Ctnnb1*△Vav1 and control LSK cells. Quantification of EdU% in LSK cells. Ctrl = 7; *Ctnnb1*△Vav1 = 7. Error bars, mean ± s.e.m. Two-tailed Student's *t* test. **l** FACS plot showing EdU incorporation into *Ctnnb1*i△Kit and control LSK cells. Quantification of EdU% in LSK cells. Ctrl = 5; *Ctnnb1*i△Kit = 4. Error bars, mean ± s.e.m. Two-tailed Student's *t* test.

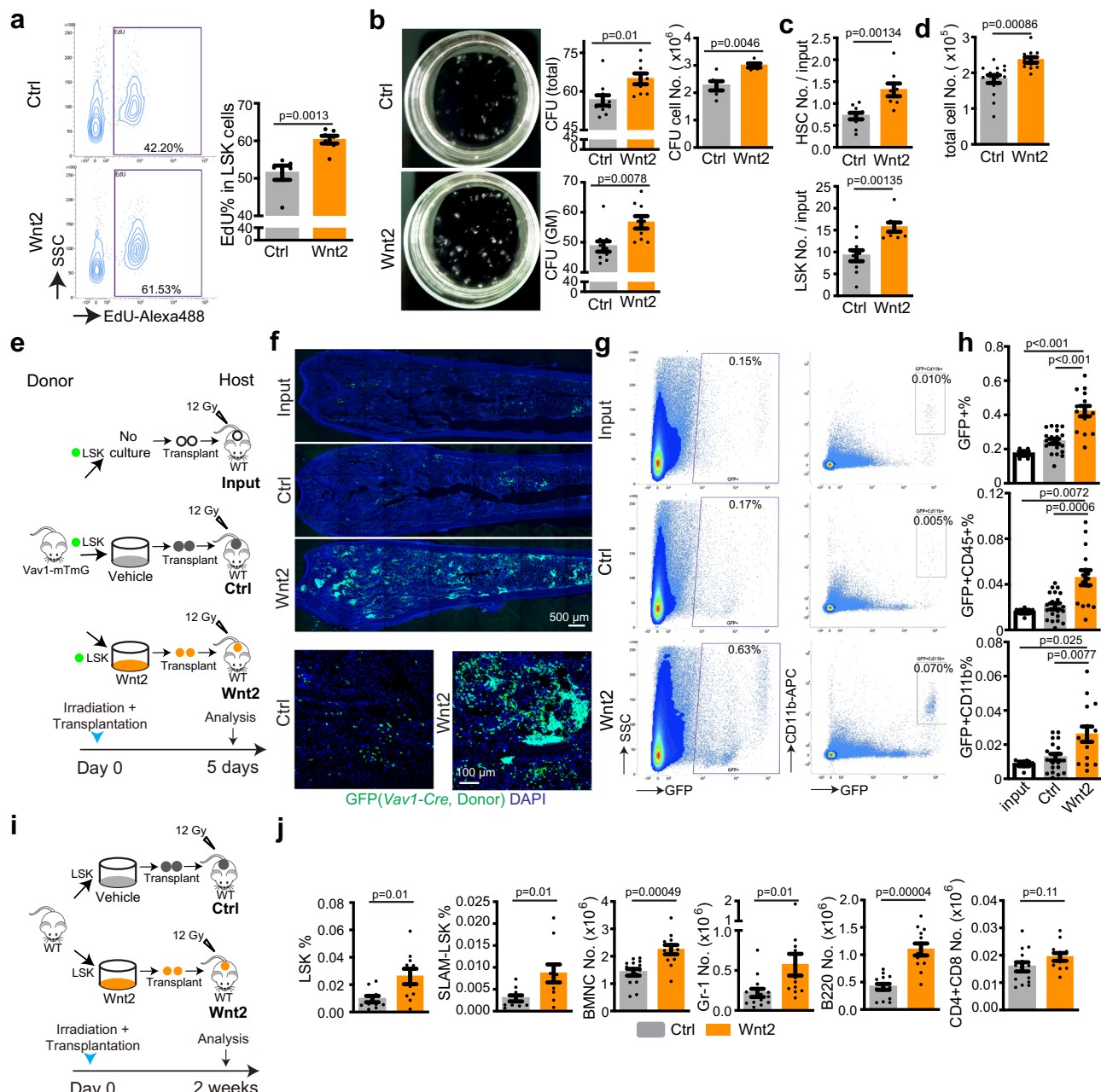

**Fig. 7 Wnt2 promotes HSPC proliferation and transplantation. a** Representative FACS plot showing transient EdU incorporation into sorted LSK cells after treatment with Wnt2 or vehicle. Quantification of EdU% in LSK cells. Ctrl = 6; Wnt2 = 7. Error bars, mean ± s.e.m. Two-tailed Student's $t$ test. **b** Representative images (8 days after seeding) and quantification of colony forming units from 600 isolated wild-type adult LSK cells. For colony count, vehicle = 10; Wnt2 = 9. For total cell number count, vehicle = 6; Wnt2 = 5. Error bars, mean ± s.e.m. Two-tailed Student's $t$ test. **c, d** Quantification of HSC number and LSK number fold change relative to adult cell input (**c**); vehicle = 10; Wnt2 = 9. Absolute cell number after Wnt2 treatment (**d**); vehicle = 15; Wnt2 = 14. Error bars, mean ± s.e.m. Two-tailed Student's $t$ test. **e** Diagram showing Wnt2-treatment of sorted adult *Vav1-mTmG* LSK cells in vitro followed by analysis at 5 days after transplantation into lethally-irradiated WT mice. **f** Representative overview and high-magnification immunostaining results showing hosts bone sections at 5 days after transplantation with Wnt2 or vehicle-pretreated GFP+ cells or fresh input cells. **g** FACS plot showing GFP+ or GFP+ CD11b+ cells from hosts at 5 days after transplantation with Wnt2 or vehicle-pretreated GFP+ cells or fresh input cells. **h** FACS based quantification of GFP+, GFP+ CD45+ and GFP+ CD11b+ cell percentage in BM at 5 days after transplantation. Input = 5; ctrl = 20; Wnt2 = 15. Error bars, mean ± s.e.m. Two-tailed Student's $t$ test. **i** Diagram showing Wnt2-treatment of sorted adult WT LSK cells in vitro followed by analysis at 2 weeks after transplantation into lethally-irradiated WT mice. **j** FACS based quantification of LSK%, SLAM-LSK% (ctrl = 10; Wnt2 = 10) as well as the number of BMNCs, Gr-1+ myeloid cells, B220+ B-lymphocytes and CD4/8+ T-lymphocytes. ctrl = 14; Wnt2 = 12. Error bars, mean ± s.e.m. Two-tailed Student's $t$ test.

Embryonic legs were dissected and immediately placed in ice-cold 4% paraformaldehyde solution and fixed under gentle agitation overnight. Leg samples were placed in 0.5 M EDTA (PH 8.0) for 1 day, dehydrated in 20% sucrose and 2% polyvinylpyrrolidone-containing PBS for 1 day, and embedded in PBS containing 20% sucrose, 8% gelatin and 2% polyvinylpyrrolidone for storage at −80 °C and cryosectioning on a Leica CM3050 cryostat using low profile blades. For

immunostaining, bone sections were rehydrated in PBS, permeabilized for 15 min in 0.5% Triton-X100 PBS solution and blocked for 30 min in PBS containing 1% BSA, 2% donkey serum, 0.3% Trion-X100 (blocking buffer) at room temperature. Sections were incubated with primary antibodies diluted in blocking buffer at 4 °C overnight. After incubation, sections were washed three times with PBS and incubated with appropriate Alexa Fluor-conjugated secondary antibodies (1:100 to

1:200, Invitrogen) diluted in blocking buffer at room temperature for 2 h. When indicated, nuclei were stained with DAPI during secondary antibody incubation. After that, sections were washed three times with PBS, mounted with Fluoromount-G (0100-01, Southern Biotech) and kept in 4 °C for imaging.

Bone sections were imaged with laser scanning confocal microscopes (Zeiss LSM780) after immunohistochemistry. Representative images without quantification were repeated using at least three biological replicates. Quantitative analysis of mutant phenotypes was done with the same microscope and identical imaging acquisition setting for mutant and control samples. Overview images of bone were automatically scanned using the tile-scan function of confocal microscope. 120 μM thick sections were used to acquire overview images showing the distribution of c-Kit+ cells in fetal BM. For overall nerve morphology in Supplementary Figs. 9c, 10h and 10o, image acquisition involved serial sections and stitching in Fiji. We used Fiji (open source; http://fiji.sc/), Volocity (PerkinElmer), Photoshop and Illustrator (Adobe) software for image processing in compliance with *Nature Communications*' guide for digital images. Original images were loaded into Volocity for brightness-contrast modifications that were applied to the whole image.

3D-reconstruction was performed in Imaris (Oxford Instruments). In general, the original confocal images were rotated and cropped in Fiji to remove surrounding muscle and cartilage before cropped images were loaded into Imaris. CD150, c-Kit, DAPI and Lin CD48 CD41 signals were designated as "dots", whereas caveolin-1, CD31, and PDGFRβ immunostaining were designated as "surface". Segmentation is mainly based on quality, size and intensity of signal and other default settings in Imaris with a comparable threshold between control and mutant. The images are captured with snapshot function in Imaris. The distance from dot to surface is calculated using the automatic "shortest distance function". These values are loaded into Graphpad Prism to perform Kolmogorov–Smirnov test for *p* value and generate graphs.

**EdU injection and analysis**. Approximate 60 mg/kg EdU (Thermo Fisher Scientific, A10044) was injected at indicated time point into pregnant dams.

For visualization of proliferating cells in sections, EdU staining (Click-it plus EdU Alexa Fluor 647 imaging kit, Invitrogen C10340) was performed after antibody staining.

For EdU+ LSK analysis by FACS, intracellular EdU staining was performed following the manufacturer's instructions (Click-it plus EdU Alexa fluor 488 flow cytometry assay kit, Invitrogen C10632).

For transient EdU labeling of LSK cells, WT LSK cells were sorted in PBS and pretreated with Wnt2 (final concentration 400 ng/ml, Abnova, H00007472-P01) or vehicle in 4 °C for 30 min before EdU (final concentration 5 μM) was added to each group and cells were incubated in 37 °C for 1 h. LSK cells were washed once, immediately fixed and subjected to intracellular EdU staining as described above.

**In vitro cell culture and Methylcellulose assay**. For in vitro culture with Wnt2 treatment, sorted LSK cells were seeded in 48 well plates (6000 cells/well) in Stemline II haematopoietic stem cell expansion medium (Sigma, S0192) containing 50 ng/ml SCF (Peprotech, 250-03) and 50 ng/ml Flt3-ligand (Peprotech, 250-31 L). Wnt2 (final concentration 400 ng/ml, Abnova, H00007472-P01) or vehicle was added to the medium when cells were seeded at day 0. At day 2 and 4, half of the medium was replaced with fresh medium together with Wnt2 (final concentration 400 ng/ml). Cells and medium were incubated at 37 °C in a 5% CO2 cell incubator and collected for analysis at day 7.

For Wnt2 treatment in combination with CFU assay, approximate 600 LSK cells (Ter119−, Gr-1−, B220−, CD3e−, c-Kit+, Sca1+) were FACS sorted and pretreated with Wnt2 (final concentration 400 ng/ml) or vehicle at 4 °C for 1 h. Next, cells were cultured in MethoCult^Tm medium (GF M3434, Stem cell technologies) in 35 mm dishes. Wnt2 (final concentration 400 ng/ml) or control vehicle were mixed with MethoCult^Tm medium during cell seeding. Cells and medium were incubated at 37 °C in a 5% CO2 cell incubator. 8 days after incubation, CFU number was counted before cells were washed into 2% FCS-PBS solution and filtered for FACS staining analysis.

Approximate 1000 Lin− c-Kit+ cells (Ter119−, Gr-1−, B220−, CD3e−, c-Kit+) from neonatal wild-type mice were FACS sorted and cultured in MethoCult^Tm medium (GF M3434, Stem cell technologies) in 35 mm dishes. Lin− c-Kit+ cells were sorted into 2% FCS-PBS with 100 μM IWR-1 (Tocris, 3532) or control vehicle. Drugs or control vehicle were mixed with MethoCult^Tm medium during cell seeding. Cells and medium were incubated at 37 °C in a 5% CO2 cell incubator. 8 days after incubation, CFU number was counted before cells were washed into 2% FCS-PBS solution and filtered for FACS staining analysis. For cell cycle analysis after staining of lineage cocktail antibodies, cells were immediately fixed using reagent A from fixation and permeabilization kit (GAS003, Invitrogen). Cells were incubated with Ki67-FITC antibody (Biolegend, 652410) for 30 min at room temperature followed by DAPI staining. Isotype control (Biolegend, 400505) was used to validate signals.

For ex vivo staining of cyclin D1, Lin− c-Kit+ cells (Ter119−, Gr-1−, B220−, CD3e−, c-Kit+) from neonatal wild-type mice were FACS sorted into precoated 8

well culture slides (Corning, 354688) and incubated at 37 °C in a 5% CO2 cell incubator with 100 μM IWR-1 or vehicle control in StemlineII haematopoietic stem cell expansion medium (Sigma, S0192) supplemented with 10% FCS. After 24 h of incubation, cells were fixed with 4% PFA for 10 min at room temperature and permeabilized with 0.15% Triton X/100 for 10 min. Cells were blocked with 1% BSA-PBS for 1 h at room temperature and incubated in 1% BSA-PBS with primary antibodies overnight at 4 °C. Slides were washed with PBS three times and incubated for 1 h at 37 °C with secondary antibodies.

**Counting of bone marrow nucleated cells (BMNCs)**. Total cell numbers were determined using a Luna 2 automated cell counter (Logos Biosystems) or cell counting plates. The number of lineage-committed cells in BM was calculated based on flow cytometry analysis with appropriate markers and BMNC counting. DAPI labeling was used for the exclusion of dead or damaged cells by FACS gating.

**Irradiation, transplantation, long-term competitive repopulation assay**. Mice were exposed to a lethal dose (12 Gy) of Gamma irradiation (137Cs, GammaCell). Bone marrow cells were transplanted 4–6 h after lethal irradiation (12 Gy).

For long-term competitive repopulating assays, CD45.1 recipient mice were lethally irradiated (12 Gy) and transplanted with all cells from two femurs of *Wls^iΔBmx* or its littermate control (CD45.2 donor) together with $5 \times 10^5$ host-derived (CD45.1 background) bone marrow cells. Host mice were sacrificed 16 weeks after transplantation to determine the level of chimerism in bone marrow and peripheral blood by flow cytometry. For transplantation of cultured *Vav1-mTmG* LSK cells into lethal irradiated WT mice, sorted *Vav1-mTmG* LSK cells were seeded in 48 well plates (9000 cells/well) in Stemline II haematopoietic stem cell expansion medium (Sigma, S0192) containing 50 ng/ml SCF and 50 ng/ml flt3-ligand. Wnt2 (final concentration 400 ng/ml) or vehicle was added to the medium during cell seeding at day 0. At day 2 and 4, half of the medium was replaced with fresh medium containing Wnt2 (final concentration 400 ng/ml). Cells and medium were incubated at 37 °C in a 5% CO2 cell incubator. On day 7, all cells from one well were collected and transplanted into one recipient wild-type mouse. Fresh 9000 *Vav1-mTmG* LSK cells without culture were used for transplantation as input group. Recipients were sacrificed after 5 days for FACS analysis or bone immunostaining.

For transplantation of WT LSK cells into lethal irradiated mice and analysis at 2 weeks, 6000 WT LSK cells were sorted and cultured in each well with Wnt2 or vehicle treated as described for cell culture experiments. On day 7, all cells from one well were collected and transplanted into one recipient wild-type mouse.

For transplantation of fetal liver, the liver of *Wls^iΔBmx* embryo and its littermate control were digested and transplanted into lethally-irradiated WT mice, which were analysed at two weeks after transplantation. In competitive repopuling assay, the liver of *Wls^iΔBmx* embryo and its littermate control were digested and mixed with a similar number of CD45.1 cells. The cell mixture was transplanted into lethally-irradiated CD45.1 mice. Chimerism was analysed at 4 months after transplantation.

**Quantification and statistical analysis**. Mice that died before the completion of experimental protocols were excluded from analysis, which was a pre-established criterion before the experiment. Embryonic mice with clear developmental delay or unexpected death were excluded from analysis. Because of data variation in different batch of mice, we normalize the data to its corresponding littermate control for statistical comparison. Thus, we only compare the fold change of mutant mice relative to the corresponding littermate controls. No statistical methods were used to predetermine sample size. Before the Student's *t* Test, samples from different groups were tested using *F* test to identify the variances between groups. *F* value less than 0.05 indicated samples have significantly different variances. Statistical data were drawn from normally distributed group. Most of samples were tested using two-tailed Student's *t* test unless stated otherwise in figure legends. *P* value less than 0.05 was considered to be statistically significant. All results are represented as mean ± s.e.m. Number of animals or cells represents biological replicates. Software packages for Leica SP5, LSM780, FACSymphony A3 Cell Analyzer and FACSuite, FACS Diva were used for data collection. Fiji, Volocity, Photochop, Illustrator, Graphpad Prism, Microsoft Excel, FACS Diva, FACSuite, FlowJo were used for data analysis.

**Reporting summary**. Further information on research design is available in the Nature Research Reporting Summary linked to this article.

## Data availability
For single cell RNA-seq analysis, the sequencing raw data and processed gene expression data, gene x cell matrices were deposited into the NCBI GEO database: GSE160741, GSE183131 and GSE152285. Mouse reference genome (mm10) weas used for gene alignment. Data supporting the findings of this study are available within the article and from the corresponding author on reasonable request. Source data are provided with this paper.

## Code availability

All the analysis scripts are available from vignettes of original software webpage of Seurat, MetaCell or iTALK. No custom code or mathematical algorithm other than variable assignment was used in this study.

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

## Acknowledgements
Y.L. was supported by Christiane Nüsslein-Volhard Stiftung, Q.C. was supported by Guangzhou Institutes of Biomedicine and Health (1103792101) and partially supported by Science and Technology Planning Project of Guangdong Province, China (2020B1212060052). Research funding was provided by the Max Planck Society, the University of Münster, the DFG (CRC 1366), the Leducq Foundation, and the European Research Council (AdG 786672, PROVEC).

## Author contributions
Y.L., Q.C. and R.H.A. designed the study, performed most experiments, interpreted the results and wrote the manuscript. H.W.J. designed, performed experiments related with scRNA-seq. H.W.J. analysed all the scRNA-seq data, interpreted the results and wrote the manuscript related with scRNA-seq. B.I.K. performed experiments. C.X. and E.W. performed experiments for adult scRNA-seq. M.S. performed FACS sorting experiments. B.Z. provided critical mouse models.

## Funding

## Competing interests
The authors declare no competing interests.
