## [Peer Review File · Nature Communications]

A specialized bone marrow microenvironment for fetal haematopoiesisReviewers' Comments:

Reviewer #1:

Remarks to the Author:

The manuscript by Liu, Cheng and Jeong describe that adult and fetal bone marrows are different in composition using single-cell RNA. The authors discuss the importance of Wnt signalling, especially Wnt2, for the expansion of HSPCs in the establishment of definitive hematopoiesis in embryonic bone marrow and the role that AECs play in this process.

The manuscript is well written although details in the description of some of the procedures are missing (see comments below) and contains a rather long section in the results that makes difficult to highlight the major findings (see specific comment below).

Overall, the manuscript presents observations of interest for the scientific community but there are major revisions that are necessary to consolidate some of the findings.

Comments:

1. Extended Figure 1a to 1e.- Authors should define what the pink staining corresponds to.
2. In line 84 the authors state that c-kit⁺ cells cannot be detected before E15.5. However, there are c-kit⁺ cells in the stainings at E15.0, although very few. Please, comment on that.
3. Extended Data Fig. 2f.- are those numbers in line with previous reports? Comment on that.
4. Extended Data Fig 2 h. What are the arrow heads labelling?
5. In line 89, the authors state "Engrafted c-Kit⁺ haematopoietic cells incorporate EdU (5-ethynyl-2-deoxyuridine)". The stainings are not clear enough to support the statements. Quantification should be included.
6. In line 90 and Extended Data Figure 2i, the authors refer to the scRNASeq without having introduced this analysis at all. Therefore at this point, it is not clear what cells are in the subset or how they were obtained. This statement should be moved to a later section.
7. Include the total number of adult and fetal cells in figure 1d legend.
8. Figure 1f.- Include the total number of cells on the plot so the absolute number can be calculated. Please include statistics to evaluate differences in OLCs-2, mitotic and eBMSC-2. Also, the authors should comment on the possibility that these differences could be caused at sampling or due to different properties of the samples.
9. In line 113 the authors state: "Other BMSC subsets vary in percentage between E18.5 and adult samples but do not show the same dichotomy as the Lepr⁺ aBMSCs and Col3a1⁺ eBMSC-1 subpopulations." What about mitotic cells? They seem to be mainly present in E18.5 samples.
10. Figures 1g, 2b, 2c and similar figures. Scales should be included for each plot.
11. Extended Data Figure 3. The authors should define the scale of the expression in panels d, f g, h and i.
12. Figure 1h.- Include a more detailed description in the legend. The authors should comment on the different architecture that can be observed in fetal and adult samples in this figure.

13. Figures 1i and 1j.- The authors do not specify what populations are involved in the comparison. Are they comparing adult cells from the aBMSC with fetal cells from the eBMSC cluster? Are on the contrary comparing all adult BMSCs with all E18.5 BMSCs? If this is the case, the analysis should be refined using clusters. Please include the number of cells in each category in the comparison in the figure legend.
14. In figure 1i, the red dots don't depict changes >2 fold change. Also, p-adjusted value should be displayed instead of p-value.
15. Fig 1j and Extended Data Fig 5l. The authors should overlay the jitter expression for individual cells or indicate the total number of cells plotted in each violin. Authors should plot the expression of these genes separated by clusters of cells. Describe the y-axis in the figure legend and include the method used to plot the violin plots in the methods section.
16. Also in this figure, Angpt1 seems to be detected in only a minority of cells which will affect the statistical analysis.
17. Line 124 reads "which implies that the niche function of LepR+ cells is not taken over by Col3a1+ eBMSCs". The authors should tone down this statement since there is no functional data to demonstrate it.
18. Extended Figures 4d and 4e.- Although the authors set the starting point of the pseudotime in the embryonic clusters there is no connection with the adult cells, the data is not a continuum but it contains discrete clusters. Therefore, pseudotime analysis should be avoided in these settings. The analysis should be repeated using algorithms (maybe like PAGA) that will give you more reliable information regarding the interrelation of the clusters.
19. Line 128 reads "In contrast to the observed differences in BMSC subpopulations, major HSPC subsets are detected at only slightly different ratios both in embryonic and adult bone samples (Extended Data Fig. 5a-5d)." Please comment on the reduction in myeloid progenitors and mild expansion of multipotent in the embryonic samples shown in Extended Data Figures 5c and 5d. Also interestingly, embryonic samples almost exclusively contain one of the 2 arms of the megakaryocytic progenitors.
20. Extended Figure 5b.- Please include scales, as mentioned before.
21. Extended Figure 5d.- As commented for Figure 1f.
22. Extended Data Fig 5e and 5k.- Authors should perform differential expression within clusters to get a better overview of the differences between adult and embryonic samples, especially in the megakaryocyte HSPC compartment and sinusoidal ECs.
23. The results of the pseudotime in Extended Data Figures 5f and 5g are unexpected. The same comments than for the previous pseudotime analysis apply but in this case, neutrophil and myeloid progenitors are expected to be closer to each other than to megakaryocytic progenitors.
24. Figures 2b, 2c, Extended Data figures 5i,5k and 5l.- Please see comments above for similar figures.
25. Figure 2h and Line 155, the authors state that "embryonic AECs express transcripts for niche factors". Please indicate the number of cells plotted for each cluster. Do adult AECs also express those factors? If so, do they express them at different levels?
26. According to Extended Data Fig. 6a, is this data obtained using solely data from fetal cells?

Authors should show the interactome corresponding to solely fetal cells. The same analysis using only adult cells would be interesting to highlight differences between fetal and adult cells. In my opinion, this figure should be included in Figure 2i since it is important for subsequent experiments. Why the authors rule out the possible interaction of SECs with HSPCs through Wnt2 shown in this figure?

27. How efficient is the ablation of Wntless in *Wls Δ Bmx* knockout mice? How specific is the recombination for AEC cells? The authors should show the staining for specific AEC marker, such as Caveolin-1 together with Extended data 6b.

28. Figure 3d.- Define how derived myeloid-, B- and T- cells were defined.

29. Figure 3e.- Include representative plot of flow analysis (and for Extended Data Fig. 6d). Also include how different populations were quantified.

30. *Wls Δ Bmx* experiments in fetal liver show that the numbers do not change; however, no transplantation experiments were performed to show that the repopulation capabilities of the fetal HSCs were unaltered.

31. Extended Data Fig. 9a. The authors could confirm the expression of Wnt signalling genes in HSPCs in the single cell RNA Seq dataset.

32. In line 225, the authors state "The sum of these results indicates that AEC-controlled Wnt/ β -catenin signalling promotes HSPC expansion in fetal BM". How can the authors be sure that ablation of Wntless does not cause a defect in colonisation of fetal BM by HSPCs in their journey from the fetal liver more than a defect in "HSPC expansion in femur", as the authors claim?

33. Figure 3m.- The values for flow data for Control and *Wls Δ Bmx* do not correspond to the data shown in the bars.

34. Overall, the reader would benefit if the authors would split the results section "AEC-derived Wnt signals regulate fetal HSC and HSPC development" into smaller sections to better highlight the different findings contained in this section.

35. Figs 4a and 4b. Include sorting strategy for populations analysed in these figures and describe the procedure used for obtaining the data in the methods section. Was the data obtained using a custom kit for BD Rhapsody Express? How many cells were analysed? Please define "Percent Expressed" and "Average Expression" in Figure legend. Is it the percentage of cells where the gene was detected? If that is the case, it will be important to indicate how many cells were surveyed as the percentage of cells expressing a gene in BM ECs is very low (5% at the most). What is the difference between "Hematopoietic cells" and "Hematopoietic cells E18.5"? The analysis should have included markers for the different EC clusters/populations so the measurements could have been split by populations to identify/confirm that AECs are the cells that are expressing Wnt2, as in Extended Data Figure 6a, AECs barely produce Wnt2 and produce mostly Wnt5. This could mean that most of the signal for Wnt2 is actually produced by SECs. Could this analysis be done from the original scRNASeq dataset?

36. Overall, it would be interesting to see if the expression patterns described in Figs 4a and 4b are also present in adult bone marrow or if this is specific for fetal bone marrow. This could be addressed from the initial scRNASeq dataset.

37. Schematic representation of Figure 4g is misleading. As it is, it looks like GFP+ cells were only used when pre-treatment with Wnt2. I assume that GFP+ LSKs were sorted and then either directly transplanted, cultured in the presence of the vehicle or cultured in the presence of Wnt2 before transplanting them. It is probably enough with changing the green colour to another one. Please, improve the description in the figure legend. I assume that foetal bone marrow cells were used. Is

that the case? Please specify in figure legend and methods section. Directly transplanted (input) results should be also shown in 4h/4i or in supplementary figures.

38. Why only myeloid cells were checked in Figure 4i? Is that because there is a strong skewing towards myeloid lineage due to culturing conditions?

39. Line 254. The authors state that Wnt2 improves transplantation efficiency. However, the methods section indicate that all cells in one well were used for transplantation. Therefore, more cells were transplanted for Wnt2 treated cells, which are likely already myeloid skewed after being cultured. The authors should transplant equal numbers of cells and look at longer time points for longer-term transplantation.

40. In methods section titled "Single cell RNA-seq analysis", authors should add the details of the sequencing data, such as ??M reads per samples, and ??M reads per cells. The authors should also include details for how the clustering was performed. Authors should indicate how many embryos and adult mice were used to obtain the data for scRNA-Seq analysis.

41. Line 541. Remove extra "in" in the sentence "Next, cells were in cultured in MethoCult medium...".

42. Indicate how DEGs were calculated in the methods section.

43. Methods section "Counting of bone marrow nucleated cells (BMNCs)". Please specify how the viability of the samples were taken into consideration for the calculations and linkage to the flow cytometry data.

44. Overall, figure legends are not very descriptive. Please include/describe all relevant information.

Reviewer #2:

Remarks to the Author:

In this interesting manuscript Liu et al., investigate the identity of the fetal microenvironment for fetal hematopoiesis and identity Wnt-producing arteriolar endothelial cells (AEC) as a niche for fetal HSPC. The authors first use scRNA-seq to examine the cellular composition of the fetal microenvironment. Consistent with previous studies they find that hematopoietic cells colonize the bone marrow after vessel invasion, and that LepR+ stromal cells –which are key component of the adult HSC niche- are absent from the fetal bone marrow. Using imaging analyses they propose that fetal HSC localize to arterioles. Analyses of their scRNAseq data suggest that AEC produce more HSC-supportive molecules than other types of endothelial cells. Selective ablation of Wntless –necessary for secretion of Wnt ligands- in AEC causes a ~20% reduction in functional HSC numbers. Conditional deletion of catenin b1 in hematopoietic cells leads to a ~4 fold reduction in phenotypical HSC and other progenitors. The authors also find that Wnt2 is the highest expressed Wnt ligand in AEC. Using a culture system they show that Wnt2 treatment promotes HSPC proliferation and colony formation. From these studies the authors conclude 1) that AEC are a niche for fetal HSC via Wnt ligand secretion and 2) "that fetal BM is fundamentally different from its adult counterpart with respect to existence of LepR+ niche cells, molecular interactions and the regulation of HSPCs". The first statement is well supported by the experimental data –except cKIT/HSC imaging analyses that lack rigor. The second statement exaggerates the novelty of the authors' findings and is not supported by the data.

Specific concerns and suggestions on how to improve the manuscript are detailed below:

Major (required)

Exaggerated claims of novelty: the central claim of the authors is that they demonstrate that the fetal microenvironment is fundamentally different from the adult microenvironment. This is based on the lack of LepR⁺ stromal cells in the fetal bone marrow and the finding that Wnt-producing AEC are a niche for fetal HSC.

1. The observation that LepR⁺ cells are not present in the fetal bone marrow has been documented in detail by various groups (see *Dev Cell*. 2014 May 12; 29(3): 340–349; *Cell Stem Cell*. 2014 Aug 7; 15(2): 154–168; *Nat Commun*. 2019; 10: 3168). The authors should cite and discuss these manuscripts when presenting and interpreting their results.

2. In the adult LepR⁺ osteoblastic stromal cells, sinusoidal endothelial cells, and arteriolar endothelial cells cooperate to maintain HSC. Coskun et al., -using *Osx* knockout mice- demonstrated that osteoblastic stromal cells are required for HSC colonization of the fetal liver (reference 19). The authors' discovery that -similar to adult AEC (*Nat Commun*. 2018 Jun 22;9(1):2449) – fetal AEC maintain fetal HSC suggest that the fetal BM microenvironment is actually similar to the adult microenvironment. Note that the Frenette lab showed that arterioles selectively maintain HSC in the fetal liver [*Science*. 2016 Jan 8; 351(6269): 176–180]. The aorta is the site of emergence of HSC in the embryo, and arterioles support both fetal liver and adult HSC. The discovery that arterioles also support fetal BM HSC –while important- seems incremental. These studies should also be considered by the authors when discussing their results.

It is possible –as the authors suggest- that there is a differential requirement for AEC-derived Wnt ligands between fetal and adult HSC. If the authors wish to make this claim then they should demonstrate that Wntless deletion in adult AEC does not perturb adult HSC.

Biological significance: The authors elegantly demonstrate that fetal AEC maintain HSC. However, the observed reduction in HSC numbers and function is modest (Fig. 3c-e). Does this reduction translate into an overall defect in fetal or adult hematopoiesis? Are mature blood cells reduced in the marrow/blood of fetal or adult *Wls Δ Bmx* mice? In other words: does perturbation of the fetal arteriolar niche result in fetal or adult hematopoietic injury? This would greatly increase the significance of the authors' findings.

Concerns regarding imaging analyses:

The authors extensively use imaging of c-Kit⁺ cells to examine HSPC colonization of the bone marrow as well as to quantify reductions in HSPC numbers in their mouse models. However –at least to this reviewer- the c-Kit staining seems inconsistent/not specific. For example, in multiple panels (e.g. Fig. 1c; Fig. 3b,f, I; Extended Data Fig. 2e-g; Extended Data Fig. 7a) c-Kit⁺ cells are uniformly distributed through the bone marrow, essentially contiguous to each other. In sharp contrast, the flow cytometry data presented by the authors (Extended Data Fig. 2j) shows that c-kit⁺ HSPC are rare. This FACS frequency is consistent with previous studies by the Hirschi (reference 19) and Weissman [*PLoS Biol*. 2004 Mar;2(3):E7] groups. The discrepancy between the authors FACS and imaging data indicates that either a) the CD117 imaging is not specific; or b) that there are numerous c-Kit⁺ cells present in the bone marrow that are not HSPC (or even hematopoietic cells). In this case the data shown in panels Fig. 1c; Fig. 3b,f, I; Extended Data Fig. 2e-g; and Extended Data Fig. 7a will need to be extensively corrected. Of note when the authors fate map c-Kit⁺ cells using *Kit-CreER* mice they find that cKit⁺ cells are rare (Extended Data Fig. 9e) further suggesting that the cKit antibody stain is not specific. I strongly encourage the authors to extensively validate the specificity of all their hematopoietic cell stains as the data presented is not convincing.

In Fig. 2d-g the authors image Lin-CD48-CD41-CD150+ HSC or Kit+ cells with arterioles at E16.5. They also measure the distance from each HSC or DAPI+ to the closest artery and compare the observed distributions. From this experiment the authors conclude that most HSC localize near arterioles. These analyses are very superficial. To the best of this reviewer knowledge this is the first time that fetal HSC are imaged in the bone marrow. This is a monumental achievement that is barely examined in the manuscript. At this developmental time only 3-5 HSC are expected to be present in the bone marrow. What is HSPC localization with other components of the microenvironment (e.g. sinusoids, or stromal cells)? Are HSC evenly distributed through the BM? Are HSC relocating away from arterioles after Wls deletion? Although the authors claim to have imaged over 50 HSC only one image is shown (Fig. 2d). More images will help the reader to better understand HSC distribution in the fetal bone marrow. Comparing the distances of HSC with DAPI+ cells is incorrect as –at this developmental time point- many DAPI+ cells are stromal or endothelial cells (as shown by the authors). HSC distances should be compared with those of randomly placed cells using the Lin/CD48/CD41+ or cKit+ signal to define the possible locations of the random cells, statistical comparisons should be provided (see Nature 526, 126–130 (2015)).

Bioinformatic analyses: The single-cell analyses were performed using well-established and rigorous methods, without evidence of clear batch effects or over-clustering, with results reasonably validated. However, the analyses presented are superficial and missing in depth.

The authors claim that “In contrast to the observed differences in BMSC subpopulations, major HSPC subsets are detected at only slightly different ratios both in embryonic and adult bone samples (Extended Data Fig. 5a-5d). At the same time, DEG analysis indicates relatively minor alterations at the transcriptional level and preserved hierarchical organization of embryonic and adult HSPCs (Extended Data Fig. 5e-5g)”. This is a major claim that directly contradicts recent studies from the Magee lab and Tan labs showing profound differences in the transcriptional program of fetal and adult HSC [Cell Stem Cell. 2020 Nov 5;27(5):732-747; Cell Rep 2019 Dec 17;29(12):4200-4211.e7] and numerous previous studies showing functional and transcriptional differences between fetal and adult HSC (see Exp Mol Med. 2013 Nov 15;45(11):e55). This needs to be addressed in detail.

Similarly the classification of the different hematopoietic populations seems superficial; why are erythroid progenitors not detected in the adult of fetal bone marrow? Can the authors provide more detailed clustering information? Are the scRNAseq subsets substantiated by FACS?

The authors show expression of HSC-supportive factors in endothelial cells. Why is this information not provided for the rest of the stromal analyses? This will be very helpful for the field as other stromal cells are expected to maintain HSC via these signals [for example the authors’ own data shows that other cells produce Wnt ligands that maintain HSC and the phenotype of the hematopoietic Ctnfb1 cKO is much stronger than the AEC Wls cKO (Fig. 3g-k)]

Minor:

- The cell-to-cluster assignments, monocle pseudo-time predictions and cluster-marker genes should be provided in supplemental Excel files to enable reproducibility.

- More detailed single-cell metrics for each sample should be provided along with the number of cell barcodes/total reads/UMI's detected or targeted for sequencing using the BD Rhapsody system. Methods for the targeted sequencing are not sufficiently detailed.

- The rationale for targeted sequencing is not clear. Does this provide better resolution for WNT signaling component genes? UMI metrics should clarify.

- In Figure 4b, how were cell populations determined? Were these aligned to the whole transcriptome data for cell assignment using a reasonable set of markers for delineating?

- In many panels of Figs 1-2 the authors use the term "sinusoidal" instead of "sinusoidal".

- To better substantiate that the *Wls1ΔBmx* phenotype is specific to the BM and not initiated in the fetal liver the authors should perform competitive HSC transplants from *Wls1ΔBmx* and control fetal liver.

- Throughout the manuscript the authors normalize HSC and hematopoietic cell numbers to that of control mice. The reason for this is unclear and complicates interpretation of the data. Please show absolute numbers per bone for each population analyzed.

- The authors frequently refer to their samples as "embryonic" even though analyses are performed at E16.5-18.5 (late fetal period).

We would like to thank all reviewers for their time, effort and constructive suggestions, which are greatly appreciated and have enabled us to improve the manuscript further. Because a substantial number of new data and analysis have been added to the manuscript, the revised version now includes 7 Figures and 13 Supplementary Figures. While a detailed point-by-point response to all comments and questions is provided below, we would like to list the additional experiments and analyses included in the revised manuscript:

- Comparison of differentially expressed genes (DEG) between E18.5 and adult at sub-cluster level for bone marrow stromal cells (BMSCs), hematopoietic stem and progenitor cells (HSPCs), and endothelial cells (ECs)
- Violin plot comparison of key genes at sub-cluster level for BMSCs, HSPCs and ECs
- Added expression scalebar for all UMAP-based feature plots of gene expression
- Comparison of interactome analysis for stromal-derived signals acting on HSPCs in E18.5 and adult
- Imaris 3D reconstruction of CD150⁺ HSCs, c-Kit⁺ HSPCs, Caveolin-1⁺ artery, CD31⁺ blood vessel, PDGFR β ⁺ stromal cell, osterix⁺ osteoprogenitor cells in fetal bone marrow (BM) to reveal location of HSCs
- Systematic analysis of the distance of HSCs, c-Kit⁺ HSPCs, DAPI-labelled cells and mature hematopoietic cells to Caveolin-1⁺ arteries based on Imaris reconstruction
- Comparison of the distance between HSCs and Caveolin-1⁺ arteries in control and *Wls*^{i Δ Bmx} fetal mice.
- Analysis of Wnt ligand expression in arterial and non-arterial ECs at E18.5 and in adult
- Hematopoietic reconstitution analysis of Wnt2-treated LSK cells at longer time points after transplantation
- Re-analysis of potential interrelations among clusters of HSPCs and BMSCs by MetaCell analysis.
- UMAP feature plots showing the expression pattern of key HSPC-supportive factors in all cells at E18.5
- Evidence for the specificity of the Bmx-CreERT2 line in BM and analysis of the knockout efficiency of *Wls*^{i Δ Bmx} fetal mice
- Analysis of HSC number and distance to Caveolin-1 arteries during early engraftment at E16.5 in *Wls*^{i Δ Bmx} fetal mice
- Inclusion of all representative FACS plots for HSPC quantification
- Transplantation efficiency analysis using liver of *Wls*^{i Δ Bmx} mice as donor
- CD45.1/CD45.2 competitive repopulation assay with *Wls*^{i Δ Bmx} liver-derived cells as CD45.2 donor
- UMAP feature plots showing expression pattern of key Wnt pathway components in HSPCs
- Validation of cKit-FITC antibody for immunostaining
- Wnt2 treatment on neonatal LSK cells for transplantation
- Transplantation efficiency and cell composition analysis when the same number of control or Wnt2-treated cells were transplanted

Reviewer #1 (Remarks to the Author):

The manuscript by Liu, Cheng and Jeong describe that adult and fetal bone marrows are different in composition using single-cell RNA. The authors discuss the importance of Wnt signalling, especially Wnt2, for the expansion of HSPCs in the establishment of definitive hematopoiesis in embryonic bone marrow and the role that AECs play in this process.

The manuscript is well written although details in the description of some of the procedures are missing (see comments below) and contains a rather long section in the results that makes difficult to highlight the major findings (see specific comment below).

Overall, the manuscript presents observations of interest for the scientific community but there are major revisions that are necessary to consolidate some of the findings.

Reply: We are very grateful for this summary of our major finding and the insightful suggestion, which have helped us to improve the manuscript. We have followed the reviewer's suggestion and have added more information to the figure legends and methods. Results are subdivided into 6 sections in the revised manuscript.

Comments:

1. Extended Figure 1a to 1e.- Authors should define what the pink staining corresponds to.

Reply: We apologize for any confusion. There is no pink channel but this color results from the co-localization of CD31 (red) and Emcn (blue) in non-arterial endothelial cells. In fetal bone marrow, most of blood vessels are both CD31+ and Emcn+, whereas the artery is red because of low or no expression of Emcn (see higher magnification image in Supplementary Fig.1a-1d).

2. In line 84 the authors state that c-kit+ cells cannot be detected before E15.5. However, there are c-kit+ cells in the stainings at E15.0, although very few. Please, comment on that.

Reply: In Fig.1a-1c and Supplementary Fig. 2a-2b, our images include bone marrow together with cartilage and surrounding muscle. From E14.5 to E15.0, as can be seen below, the vascularization of the femoral cartilage template is beginning and marrow formation is not yet initiated (Reviewer 1 Figure 1, left panel). The formation of primary bone marrow can be detected from E15.5 (Reviewer 1 Figure 1, right panel).

Scarce c-Kit+ cells are detected in cartilage and surrounding muscle, whereas we do not see signal in the area giving rise to bone marrow.

Reviewer 1 Figure 1 Structure of primary bone marrow and surrounding tissues

3. Extended Data Fig. 2f.- are those numbers in line with previous reports? Comment on that.

Reply: Previous work has reported CFU ability and CD45.2/CD45.1 competitive repopulation ability of embryonic bone marrow (Figure 2A and 2C in PMID: 25310984, or see Reviewer 1 Figure 2).

Consistent with our finding, there is no competitive repopulation ability before E16.5 (Figure 2C in PMID: 25310984). CFU ability gradually increases from E16.5 to E18.5 (Reviewer 1 Figure 2, left panel), which shows a similar trend as our quantification of the c-Kit⁺ area (Supplementary Fig. 2f).

This indicates that our data is consistent with previous reports.

Reviewer 1 Figure 2 CFU and engraftment ability of embryonic bone marrow (PMID: 25310984)

4. Extended Data Fig 2 h. What are the arrow heads labelling?

Reply: The arrowheads indicate co-localization of c-Kit and EdU signal. See also next question.

5. In line 89, the authors state “Engrafted c-Kit⁺ haematopoietic cells incorporate EdU (5-ethynyl-2-deoxyuridine)”. The stainings are not clear enough to support the statements. Quantification should be included.

Reply: Agree. While we have kept the overview image to show c-Kit and EdU signal distribution in BM (Supplementary Fig. 2h, upper panel), we replaced the three image panels at the bottom with versions showing higher magnification (Supplementary Fig. 2h, lower panels). This change makes it easier to identify the double positive cells.

We quantify the percentage of EdU⁺ signal in c-Kit cells by immunostaining and FACS. Consistently, the two approaches show that $33.0\% \pm 3.6\%$ and $32.0\% \pm 3.7\%$ of c-Kit cells are EdU⁺, respectively (Supplementary Fig. 2h).

6. In line 90 and Extended Data Figure 2i, the authors refer to the scRNASeq without having introduced this analysis at all. Therefore at this point, it is not clear what cells are in the subset or how they were obtained. This statement should be moved to a later section.

Reply: We agree. Because the EdU immunostaining and FACS provide clear evidence for HSPC proliferation in fetal BM, we no longer use the scRNA-seq results for this purpose.

At the same time, we have extended the analysis of the scRNA-seq data for HSPCs in the revised manuscript.

7. Include the total number of adult and fetal cells in figure 1d legend.

Reply: Agree. We now provide the number of all cells (Supplementary Fig. 3e), HSPCs (Fig. 1g), BMSCs (Fig. 2a) and ECs (Fig. 3a) in the corresponding graphs.

8. Figure 1f.- Include the total number of cells on the plot so the absolute number can be calculated. Please include statistics to evaluate differences in OLCs-2, mitotic and eBMSC-2. Also, the authors should comment on the possibility that these differences could be caused at sampling or due to different properties of the samples.

Reply: As suggested by the reviewer, we now provide the number of adult and fetal BMSC cells in Fig. 2a.

In the revised manuscript, we also include statistics to evaluate differences for all 6 clusters of fetal and adult BMSCs (Fig. 2f). Except eBMSC-2, all clusters show a large number of differentially expressed genes (DEGs) between E18.5 and adult.

The quantification of differences among all BMSCs is displayed using Pearson's analysis (see Reviewer 1 Figure 3 below). These differences are very unlikely to be induced by sampling. To rule out the possibility of sampling differences and variation among mice, we had pooled 5 and 4 different E18.5 and adult animals, respectively.

Reviewer 1 Figure 3 Pearson's analysis of differences for all BMSC sub-clusters

9. In line 113 the authors state: "Other BMSC subsets vary in percentage between E18.5 and adult samples but do not show the same dichotomy as the Lepr+ aBMSCs and Col3a1+ eBMSC-1 subpopulations." What about mitotic cells? They seem to be mainly present in E18.5 samples.

Reply: We agree that mitotic cells are mainly present in E18.5 samples, which probably reflects developmental growth and not so much a difference in the cellular composition of marrow stroma. We also focus on the Lepr+ aBMSC and Col3a1+ eBMSC-1 subpopulations because they are the major BMSC populations (>55%) in adult and at E18.5, respectively.

10. Figures 1g, 2b, 2c and similar figures. Scales should be included for each plot.

Reply: Agree. Scales have been added to these plots.

11. Extended Data Figure 3. The authors should define the scale of the expression in panels d, f g, h and i.

Reply: Agree. Scales have been added to these plots.

12. Figure 1h.- Include a more detailed description in the legend. The authors should comment on the different architecture that can be observed in fetal and adult samples in this figure.

Reply: As requested, we have added a more detailed description in the figure legend (now, Fig. 2c):

“Representative projected confocal images showing the number of Lepr-Cre-labelled perivascular cells in adult or E18.5 R26-mTmG Cre reporter bone marrow. Lepr-Cre-controlled GFP expression predominantly labels sinusoidal vessel-associated (reticular) cells in adult BM, whereas GFP+ cells are very sparse at E18.5. ECs, Emcn (red).”

In this context, it should be also noted that the newly formed vasculature in embryonic bone looks rather irregular in comparison to postnatal stages (see Figure 1A of PMID: 28218908 or Reviewer 1 Figure 4). This previous publication is also the reason why do not comment very much on general vascular architecture in the current manuscript.

Reviewer 1 Figure 4 Vascular development in long bone (PMID: 28218908)

13. Figures 1i and 1j.- The authors do not specify what populations are involved in the comparison. Are they comparing adult cells from the aBMSC with fetal cells from the eBMSC cluster? Are on the contrary comparing all adult BMSCs with all E18.5 BMSCs? If this is the case, the analysis should be refined using clusters. Please include the number of cells in each category in the comparison in the figure legend.

Reply: We agree. In Figure 1i and 1j (now, Fig. 2d and 2e), we compare “all adult BMSCs” with “all E18.5 BMSCs”. In the revised manuscript, a label indicating “All BMSCs (E18.5 vs adult)” has been added to Fig. 2d.

Following the reviewer’s constructive suggestion, we have performed DEG analysis for different cell clusters (Figure 2f). For example, 419 DEGs distinguish the “adult eBMSC-1” and “E18.5 eBMSC-1” populations. The number of cells in each category for comparison is now shown in the right panel of Fig. 2a.

14. In figure 1i, the red dots don’t depict changes >2 fold change. Also, p-adjusted value should be displayed instead of p-value.

Reply: Thank you for alerting us to this mistake, which has been corrected in the revised manuscript.

We now show p-adjusted value and red dots represent genes with (1) p-adjusted value < 0.05 and (2) Log₂ fold change (Log₂ FC > 0.5) or (Log₂ FC < -0.5).

15. Fig 1j and Extended Data Fig 5l. The authors should overlay the jitter expression for individual cells or indicate the total number of cells plotted in each violin. Authors should

plot the expression of these genes separated by clusters of cells. Describe the y-axis in the figure legend and include the method used to plot the violin plots in the methods section.

Reply: Agree. Following the reviewer's constructive suggestion, we are now showing the jittered data points for individual cells in each violin plot in Fig. 1J (now Fig. 2e) and Extended Data Fig. 5I (now Fig. 3d). We also plot the expression of these genes for separated clusters of cells (Fig. 2g, Fig. 4g and Supplementary Fig. 6a).

A sentence describing the method used to plot the violin plots has been added to the section "Single cell RNA-seq data analysis" in the Methods: "*We used "VlnPlot" function of Seurat package for the plots which plotting the expression level of each gene with the log normalized value by default.*"

We have also added the relevant description to the legends of Fig. 2e and Fig. 3d.

16. Also in this figure, *Angpt1* seems to be detected in only a minority of cells which will affect the statistical analysis.

Reply: Agree. The expression of *Angpt1* in each cluster at E18.5 and in adult is shown in Fig. 2g and Supplementary Fig. 4d. In the comparison of BMSC subclusters, *Angpt1* expression in E18.5 aBMSCs is substantially lower than in adult aBMSCs (Fig. 2g). This is fully consistent with our conclusion that *Angpt1* expression in E18.5 BMSCs is significantly lower. Moreover, *Angpt1* expression in E18.5 bone is not taken over by another BMSC subpopulation.

17. Line 124 reads "which implies that the niche function of *LepR*⁺ cells is not taken over by *Col3a1*⁺ eBMSCs". The authors should tone down this statement since there is no functional data to demonstrate it.

Reply: Agree. We have rephrased this sentence, which now states "*which implies that the secretion of these niche factors by *LepR*⁺ cells is not taken over by *Col3a1*⁺ eBMSCs*". This sentence is more specific and supported by our scRNA-seq data (Fig. 2e and 2g).

18. Extended Figures 4d and 4e.- Although the authors set the starting point of the pseudotime in the embryonic clusters there is no connection with the adult cells, the data is not a continuum but it contains discrete clusters. Therefore, pseudotime analysis should be avoided in these settings. The analysis should be repeated using algorithms (maybe like PAGA) that will give you more reliable information regarding the interrelation of the clusters.

Reply: Agree. We have now used the Metacell algorithm (Baran et al, Genome Biology 20:206 (2019), <https://tanaylab.github.io/metacell/>) to re-analyze the interrelation of cell clusters (Supplementary Fig. 4g).

Metacell facilitates analysis of single cell RNA-seq data by computing partitions of a cell similarity graph into small homogeneous groups of cells defined as metacells. The derived

metacells are then used for building different representations of the data depicting the interrelation of the cell clusters in matrix or 2D graph visualization.

19. Line 128 reads “In contrast to the observed differences in BMSC subpopulations, major HSPC subsets are detected at only slightly different ratios both in embryonic and adult bone samples (Extended Data Fig. 5a-5d).” Please comment on the reduction in myeloid progenitors and mild expansion of multipotent in the embryonic samples shown in Extended Data Figures 5c and 5d. Also interestingly, embryonic samples almost exclusively contain one of the 2 arms of the megakaryocytic progenitors.

Reply: From our point of view, one plausible explanation is that these changes in the percentage of myeloid progenitors and multipotent HSPCs reflect the fact that the hematopoietic hierarchy is not yet fully and stably established at E18.5. For example, the hematopoietic system takes several months to reconstitute and stabilize after irradiation and BM transplantation. Although we are analyzing development and not hematopoietic reconstitution, our time point of analysis (E18.5) is only two days after the initial engraftment of HSCs and c-Kit⁺ HSPCs at E16.5 (Fig. 4a-4e).

At E18.5, HSCs may not yet differentiate into oligopotent HSPCs at the same rate as in adult mice. The scRNA-seq data is consistent with our FACS results. The percentage of LSK cells at E18.5 is approximate $0.084\% \pm 0.017\%$ (Mean \pm SEM) while the LSK% is widely known to be about 0.05% in adult BM. The percentage of myeloid progenitors at E18.5 and adult are $0.25\% \pm 0.03\%$ (Mean \pm SEM) and $0.64\% \pm 0.03\%$ (Mean \pm SEM), respectively. Therefore, the variation of cell percentage may reflect the ongoing development towards a stable hematopoietic hierarchy.

Reviewer 1 Figure 5 Megakaryocytic progenitors at E18.5 and adult

Regarding the megakaryocytic progenitors, we find Itga6 (Integrin alpha6/CD49f, encoded by *Itga6*) is enriched in the left arm of this cluster (Reviewer 1 Figure 5). FACS analysis suggests that about 2.12% of Lin⁻ cells are CD41⁺Itga6⁺ in adult BM but only 0.1% of Lin⁻ cells are CD41⁺Itga6⁺ at E18.5. These results may indicate two populations of megakaryocytic progenitors, which can be distinguished by expression of Itga6. CD41⁺Itga6⁺ progenitors may not yet have fully formed at E18.5 in BM. We fully agree with the reviewer’s opinion that this is a very interesting question, which deserves further investigation in a future project.

20. Extended Figure 5b.- Please include scales, as mentioned before.

Reply: Agree. Scales have been added.

21. Extended Figure 5d.- As commented for Figure 1f.

Reply: Agree. Information showing the number of cells in each cluster has been added.

22. Extended Data Fig 5e and 5k.- Authors should perform differential expression within clusters to get a better overview of the differences between adult and embryonic samples, especially in the megakaryocyte HSPC compartment and sinusoidal ECs.

Reply: Agree. We have modified our analysis of DEGs to compare adult and E18.5 samples in clusters (Fig. 1h, 2f and 3c).

The E18.5 and adult megakaryocytic HSPCs compartment shows a large difference with 2292 DEGs, which may relate to the existence of Lin-CD41+Itga6+ megakaryocytic progenitors as discussed under “Question 19” (Reviewer 1 Figure 5).

The E18.5 and adult sinusoidal EC compartment shows 209 DEGs when a comparison is performed within this cluster (Fig. 3c).

23. The results of the pseudotime in Extended Data Figures 5f and 5g are unexpected. The same comments than for the previous pseudotime analysis apply but in this case, neutrophil and myeloid progenitors are expected to be closer to each other than to megakaryocytic progenitors.

Reply: Agree. Similar to “Question 18”, we replaced the old pseudotime analysis by Metacell algorithms (<https://tanaylab.github.io/metacell/>) to re-analyze the interrelation of HSPC clusters. Consistent with reviewer’s view, neutrophil and myeloid progenitors are closer to each other in comparison to megakaryocytic progenitors (Supplementary Fig. 3i).

24. Figures 2b, 2c, Extended Data figures 5i,5k and 5l.- Please see comments above for similar figures.

Reply: Agree. We have added scales for all UMAP feature plots. All violin plots are now overlaid with data points for individual cells. For all DEG analysis, we compared E18.5 and adult in each cluster instead of the whole cell population.

25. Figure 2h and Line 155, the authors state that “embryonic AECs express transcripts for niche factors”. Please indicate the number of cells plotted for each cluster. Do adult AECs also express those factors? If so, do they express them at different levels?

Reply: Agree. All violin plots are now overlaid with jitter points. The number of cells can be seen in Fig. 4g (E18.5) and Supplementary Fig. 6a (adult).

Adult AECs also express some of those factors, but expression of *Jag1*, *Jag2*, and *Tgfb2* is significantly lower than at E18.5. The expression level of other factors is not substantially different in E18.5 and adult AECs.

26. According to Extended Data Fig. 6a, is this data obtained using solely data from fetal cells? Authors should show the interactome corresponding to solely fetal cells. The same analysis using only adult cells would be interesting to highlight differences between fetal and adult cells. In my opinion, this figure should be included in Figure 2i since it is important for subsequent experiments. Why the authors rule out the possible interaction of SECs with HSPCs through *Wnt2* shown in this figure?

Reply: Agree. Results in Extended data Fig.6a (now removed from manuscript) was based on data from both embryonic and adult samples. To have more robust and reliable results, we have now performed targeted scRNA-seq analysis for adult BM and compared the resulting data with our targeted analysis for E18.5 BM. These new analyses are included in Fig. 5a-5b and Supplementary Fig. 7a-7b.

The interactome analyses with solely fetal cells and solely adult cells are displayed in Fig. 5b. Overall, *Wnt* ligand expression and potential interactions are substantially different between E18.5 and adult (Fig. 5a-5b). Expression of Frizzled receptors is also different but the changes are relatively moderate in comparison to E18.5 (Supplementary Fig. 7b).

As our new targeted scRNA-seq data distinguishes between arterial and non-arterial ECs (Fig. 5a), it is evident that E18.5 AECs are a major source of *Wnt2*. This is consistent with the localization of early HSCs/HSPCs in proximity of fetal arteries. Furthermore, as we show in our answer to “Question 27”, recombination in *Bmx-CreERT2* transgenic mice is very specific for arterial ECs. While our study has focused on AEC-derived *Wnt* signals, this obviously does not rule out important contributions of SECs or other stromal cell populations.

We have therefore added the following sentence to the Discussion “*While our findings do not rule out potential contributions of sinusoidal vessels, osteoblastic cells and other stromal cell populations, we show that *LepR*⁺ reticular cells are absent in fetal BM and that embryonic HSCs and HSPCs are associated with the artery in the developing femur.*”

27. How efficient is the ablation of *Wntless* in *Wls Δ Bmx* knockout mice? How specific is the recombination for AEC cells? The authors should show the staining for specific AEC marker, such as Caveolin-1 together with Extended data 6b.

Reply: The GFP reporter signal reflecting *Bmx-CreERT2* activity is very specific for AECs in BM and, as was shown before, the same applies to arteries in many other organs (Figure S4A-S4C in PMID: 31761723; Fig 5 in PMID: 29934585). We show that GFP signal co-localizes with Caveolin-1 in both in the trochanter artery (Supplementary Fig. 7d) and arteries inside BM (Supplementary Fig. 7e).

Regarding recombination efficiency, it should be noted that *Bmx-CreERT2* works well in the endothelium of larger and therefore more mature arteries but typically spares the most distal

and newly formed arterioles. Because the number of AECs in embryonic femur is very low, we have isolated AECs from whole embryos for qPCR analysis. According to these results, the ablation efficiency is about 60% by qPCR (Supplementary Fig.7f).

28. Figure 3d.- Define how derived myeloid-, B- and T- cells were defined.

Reply: Myeloid, B, T cells were defined as CD11b+, B220+ and CD4/CD8+ cells respectively. This information is also found in the figure legends in several instances.

Wls^{iΔBmx} and its corresponding control mice were in the CD45.2 background while their competitor cells were CD45.1 (Fig. 5e). CD45.2 or CD45.1 identities of hematopoietic cells were determined by FACS. We performed the analysis at 4 months after transplantation so that hematopoietic cells were derived from transplanted HSCs.

29. Figure 3e.- Include representative plot of flow analysis (and for Extended Data Fig. 6d). Also include how different populations were quantified.

Reply: We are now showing representative FACS plots in Supplementary Fig. 7h and Supplementary Fig. 8a.

The absolute number of cells was obtained by FACS based on gating that is also shown in Supplementary Fig. 7h and Supplementary Fig. 8a. Absolute numbers were normalized to corresponding littermate control (set to 1) to obtain the fold change, which is displayed in the corresponding graphs.

We have updated the relevant figure legends accordingly.

30. *Wls^{iΔBmx}* experiments in fetal liver show that the numbers do not change; however, no transplantation experiments were performed to show that the repopulation capabilities of the fetal HSCs were unaltered.

Reply: Agree. We have added new data addressing this point. Fetal livers from *Wls^{iΔBmx}* mutants and littermate controls were used as donor and transplanted into lethally-irradiated WT mice to analyze transplantation efficiency at two weeks after transplantation. The number of BMNC, Gr-1 myeloid cell, B220 B-lymphocyte, CD4+CD8 T-lymphocyte and LSK%, HSC% were not significantly different between *Wls^{iΔBmx}* and control (Supplementary Fig. 8g-8h).

Furthermore, we have used *Wls^{iΔBmx}* and littermate control fetal liver as CD45.2 donors in combination with CD45.1 donor cells for transplantation into lethally-irradiated CD45.1 mice. Chimerism for myeloid cell, B cell and T cell populations was analyzed at 4 months after transplantation and no significant differences between mutant and control donors were observed (Supplementary Fig. 8e-8f).

31. Extended Data Fig. 9a. The authors could confirm the expression of Wnt signalling genes in HSPCs in the single cell RNA Seq dataset.

Reply: Agree. Consistent with our other results, our scRNA-seq data indicates expression of multiple components of the canonical Wnt signaling pathway in multipotent HSPCs. This includes components of Wnt receptors (LRP5, LRP6), signaling genes (Ctnnb1, Gsk3b, Apc) and downstream targets (Myc, Ccnd1) (Supplementary Fig.11b).

32. In line 225, the authors state “The sum of these results indicates that AEC-controlled Wnt/ β -catenin signalling promotes HSPC expansion in fetal BM”. How can the authors be sure that ablation of Wntless does not cause a defect in colonisation of fetal BM by HSPCs in their journey from the fetal liver more than a defect in “HSPC expansion in femur”, as the authors claim?

Reply: As discussed above, we have shown that liver hematopoiesis is not significantly different in *Wls^{iΔBmx}* and littermate control embryos. While it is technically not feasible to test whether the same amount of HSPCs is found in the embryonic circulation, we have analyzed the number of CD150+Lin-CD48-CD41- HSC and c-Kit+ HSPC number in femur at E16.5. At this very early stage of BM colonization by haematopoietic cells, the number of CD150+ HSCs and c-Kit+ HSPCs is already significantly lower in *Wls^{iΔBmx}* mutants (Supplementary Fig. 7i). This might potentially reflect that AEC-derived Wnt signals also control HSPC engraftment in embryonic bone.

We mention these points in the third paragraph of the Discussion: “*AEC-derived Wnt enhances HSPC expansion and proliferation, but, as we see a reduction of these cells already in the E16.5 femur, additional roles in the circulation, entry or engraftment of fetal HSPCs should not be ruled out.*”

33. Figure 3m.- The values for flow data for Control and *Wls^{iΔBmx}* do not correspond to the data shown in the bars.

Reply: We thank the reviewer for pointing out this mistake. While the quantification was correct as shown, we have now inserted the correct FACS plots (Fig. 6j).

34. Overall, the reader would benefit if the authors would split the results section “AEC-derived Wnt signals regulate fetal HSC and HSPC development” into smaller sections to better highlight the different findings contained in this section.

Reply: Agree. This has been done.

35. Figs 4a and 4b. Include sorting strategy for populations analysed in these figures and describe the procedure used for obtaining the data in the methods section. Was the data obtained using a custom kit for BD Rhapsody Express? How many cells were analysed? Please define “Percent Expressed” and “Average Expression” in Figure legend. Is it the percentage of cells where the gene was detected? If that is the case, it will be important to indicate how many cells were surveyed as the percentage of cells expressing a gene in BM ECs is very low (5% at the most). What is the difference between “Hematopoietic cells” and “Hematopoietic cells E18.5”? The analysis should have included markers for the different EC clusters/populations so the measurements could have been split by populations to

identify/confirm that AECs are the cells that are expressing Wnt2, as in Extended Data Figure 6a, AECs barely produce Wnt2 and produce mostly Wnt5. This could mean that most of the signal for Wnt2 is actually produced by SECs. Could this analysis be done from the original scRNASeq dataset?

Reply: We have added a new section “Targeted single-cell RNA-seq analysis for Wnt signaling pathway” to the Methods with detailed information.

The data was obtained using a custom targeted gene panel on the BD Rhapsody platform.

For adult, we recovered 7,948 Lin⁻ cells. For E18.5, we performed this targeted scRNA-seq separately for Lin⁻ cells and ECs in a total of 2 runs. After the sequencing and data QC, we recovered 12,884 Lin⁻ cells and 4,043 ECs, respectively. We had to add ECs for the 2nd run of analysis because the number of ECs in the Lin⁻ batch was insufficient for analysis.

“Percent expressed” means percentage of cells where the gene was detected. “Average expression” means average expression level of the gene (scaled data) in each cluster. We have improved the description in the figure legends.

In the original submission, we had divided hematopoietic cells into two different clusters, both for E18.5, using unsupervised clustering. We have now removed the previous analysis and combine all E18.5 data together with newly added adult data to address questions 26, 35 and 36.

In revised manuscript, we followed the reviewer’s suggestion to use markers to define different population of cells (Supplementary Fig. 7a), which indicates that different cell types were successfully clustered with the custom panel scRNA-seq data. Both arterial and non-arterial ECs show high expression of Kdr (VEGFR2), a marker for ECs (Supplementary Fig. 3b, cluster 5 for all cells). Arterial ECs show enrichment of markers such as Gja4 (Supplementary Fig.7 a and Supplementary Fig.5b), which show much lower expression in non-arterial ECs.

Based on our new analysis, arterial but not other ECs are a major source of Wnt ligands in the BM vasculature both at E18.5 and in adult (Fig. 5a). Wnt2 is the major ligand detected at E18.5 but shows comparably low expression in adult AECs (Fig. 5a).

Given that Wnt ligand transcripts are not sufficiently covered by whole transcriptome analysis, we had to utilize a custom scRNA-seq panel that was specifically designed for the Wnt pathway. Targeted sequencing was used both for E18.5 and adult BM so that results are comparable.

36. Overall, it would be interesting to see if the expression patterns described in Figs 4a and 4b are also present in adult bone marrow or if this is specific for fetal bone marrow. This could be addressed from the initial scRNASeq dataset.

Reply: Agree. For the reasons discussed above, we have used targeted scRNA-seq for this purpose. Overall, the expression of Frizzled receptors shows more moderate changes in different BM cell populations between E18.5 and adult (Supplementary Fig. 7b), whereas we found bigger differences for Wnt ligand expression by ECs (Fig.5a). As a consequence, the

whole interactome analysis also shows substantial differences between E18.5 and adult (Fig. 5b).

37. Schematic representation of Figure 4g is misleading. As it is, it looks like GFP+ cells were only used when pre-treatment with Wnt2. I assume that GFP+ LSKs were sorted and then either directly transplanted, cultured in the presence of the vehicle or cultured in the presence of Wnt2 before transplanting them. It is probably enough with changing the green colour to another one. Please, improve the description in the figure legend. I assume that foetal bone marrow cells were used. Is that the case? Please specify in figure legend and methods section. Directly transplanted (input) results should be also shown in 4h/4i or in supplementary figures.

Reply: Agree. We are sorry for the confusion and have addressed these issues.

“I assume that GFP+ LSKs were sorted and then either directly transplanted, cultured in the presence of the vehicle or cultured in the presence of Wnt2 before transplanting them.”

This assumption is correct. We now indicate (1) the Wnt2 treatment with the color “orange” to avoid misunderstanding and (2) modified the schematic representation (Fig. 7e).

“I assume that foetal bone marrow cells were used.”

We used adult LSK cells for the Wnt2 *in vitro* treatment experiment because the number of LSK cells in fetal BM is very limited and insufficient for *in vitro* culture and transplantation into irradiated recipients. But, as discussed above, there is no major change in Frizzled receptor expression in E18.5 vs adult HSPCs. Moreover, we confirmed that recombinant Wnt2 is also effective for LSK cells derived from neonatal pups in both *in vitro* culture and transplantation experiments (Supplementary Fig. 13b-13d).

The directly transplanted (input) results are shown in Fig. 7f-7h.

38. Why only myeloid cells were checked in Figure 4i? Is that because there is a strong skewing towards myeloid lineage due to culturing conditions?

Reply: After transplantation, myeloid differentiation is fastest. At 5 days after transplantation, the number of newly generated lymphocytes is very limited and we therefore only checked myeloid cells. We choose 5 days after transplantation for our analysis because there are too many GFP+ cells at later stages, which prevents a meaningful comparison by immunostaining of sections (Fig. 7f). In the revised manuscript, we added experiments with one later time point (2 weeks after transplantation) where all cells in the culture dish were transplanted into lethally irradiated mice (Fig. 7i). At 2 weeks after transplantation, the number of both Gr-1 myeloid cells and B220 lymphocyte is increased both when neonatal or adult LSK cells were treated with Wnt2 (Fig. 7i-7j and Supplementary Fig. 13d).

To check whether there is skewing towards the myeloid lineage, we checked the percentage of LSK cells with expression of CD16/32 (FcR II and III, marker of myeloid progenitors) and

found no significant increase (Reviewer 1 Figure 6). But we detect a higher percentage of CD11b cells after Wnt2 treatment *in vitro* (Reviewer 1 Figure 6).

Reviewer 1 Figure 6 Wnt2 treatment induced marker expression *in vitro*

To investigate whether Wnt2 treatment will induce skewing towards the myeloid lineage during transplantation, we transplanted the same number of cells from culture wells (not all cells from each well; see Question 39). In this setting, BMNC number is not significantly increased after Wnt2 treatment, but we detect a higher number of Gr-1 myeloid cells as well as lower number B220 B-lymphocytes and CD4+ CD8+ T-lymphocytes (Reviewer 1 Figure 7; see Question 39). This suggests that Wnt2 can induce myeloid lineage skewing during transplantation. These results do not contradict our other transplantation experiments (Fig.7i-7j and Supplementary Fig.13d), which show that the percentage of donor multipotent stem/progenitor cells is increased after Wnt2 treatment prior to transplantation.

39. Line 254. The authors state that Wnt2 improves transplantation efficiency. However, the methods section indicate that all cells in one well were used for transplantation. Therefore, more cells were transplanted for Wnt2 treated cells, which are likely already myeloid skewed after being cultured. The authors should transplant equal numbers of cells and look at longer time points for longer-term transplantation.

Reply: It is correct that treatment with Wnt2 *ex vivo/in vitro* increases the number of HSPCs and colony formation, which is further supported by the transplantation experiments with all cells per well. As suggested by the reviewer, we also transplanted the same number of Wnt2 treated cells and performed the analysis at 3 weeks after transplantation. In this setting, we cannot detect a significant increase in BMNC number for Wnt2-treated donor cells, whereas we detect myeloid skewing as mention in our reply to Question 38 (Reviewer 1 Figure 7a).

Similarly, when the same number of control or Wnt2-treated cells were transplanted and analyzed at 4 months after transplantation, the percentage of LSK cells and HSCs in recipient mice showed no significant differences (Reviewer 1 Figure 7b).

These results are not unexpected because treatment with recombinant Wnt2 induces an increase in HSPC number, whereas we have no evidence that it enhances the stem cell properties and repopulation potential of individual cells.

In this context, it should be also mentioned that previous publications investigating *in vitro* expansion of HSPCs have also transplanted all cells from a treated well into irradiated mice (e.g. Fig. 3 of PMID: 30988422) because the transplantation is used as a biological readout for the effect of the *in vitro* treatment.

40. In methods section titled "Single cell RNA-seq analysis", authors should add the details of the sequencing data, such as ??M reads per samples, and ??M reads per cells. The authors should also include details for how the clustering was performed. Authors should indicate how many embryos and adult mice were used to obtain the data for scRNA-Seq analysis.

Reply: Agree. We apologize for the insufficient information of the sequencing and data analysis. We have updated the Methods accordingly. For whole transcriptome analysis, we performed sequencing with an Illumina NextSeq500 High-output (up to 400M reads) kit (150 cycles) for each sample aiming at a sequencing depth of >20K reads per cell, approximately. After the sequencing, data QC and genome alignment, finally we recovered 16,171 and 10,203 reads per cell for adult and embryonic samples, respectively. The sequencing saturation percentages were 87.69% and 80.06%.

The same significant principal components used for UMAP non-linear dimensional reduction were also used for the unsupervised hierarchical clustering analysis using FindClusters function in Seurat. We tested different resolutions between 0.1 ~ 0.9 and used clustree R package to select the most stable and most relevant resolution. Cellular identity of each cluster was determined by finding cluster-specific marker genes using FindAllMarkers function with minimum fraction of cells expressing the gene over 25% (min.pct=0.25).

For adult targeted scRNA-seq, 4 adult mice were sacrificed and we obtained 7,948 Lin- cells. At E18.5, 8 embryos were used to obtain Lin- cells and 8 embryos were used to FACS sort CD31+ ECs. The Lin- cells and ECs were mixed and loaded for targeted scRNA-seq at E18.5. After the sequencing and data QC, we recovered 12,884 Lin- cells and 4,043 ECs, respectively. We had to add ECs from 2nd runs of analysis because the number of ECs in the first batch of E18.5 Lin- cells was insufficient.

41. Line 541. Remove extra "in" in the sentence "Next, cells were in cultured in MethoCult medium...".

Reply: Thank you for alerting us to this mistake.

42. Indicate how DEGs were calculated in the methods section.

Reply: Differentially expressed genes or cluster-specific marker genes were identified by using the non-parametric Wilcoxon rank sum test with FindMarkers or FindAllMarkers functions of Seurat package, respectively. We used default options for the analyses if not specified otherwise. We have added this description to the Methods.

43. Methods section “Counting of bone marrow nucleated cells (BMNCs)”. Please specify how the viability of the samples were taken into consideration for the calculations and linkage to the flow cytometry data.

Reply: In general, we add DAPI to deplete dead cells by gating in our flow cytometry analysis (Reviewer 1 Figure 8).

We also evaluated the viability of RBC-depleted cells. The average DAPI- cell percentage in whole BMNC is 98.46%±0.23% (Mean±SEM).

Reviewer 1 Figure 10 Viability of BMNC

44. Overall, figure legends are not very descriptive. Please include/describe all relevant information.

Reply: We have followed reviewer’s suggestion and updated the figure legends. As in every paper, we have to make a compromise between the length of the legend and the amount of detail provided.

Reviewer #2 (Remarks to the Author):

In this interesting manuscript Liu et al., investigate the identity of the fetal microenvironment for fetal hematopoiesis and identify Wnt-producing arteriolar endothelial cells (AEC) as a niche for fetal HSPC. The authors first use scRNA-seq to examine the cellular composition of the fetal microenvironment. Consistent with previous studies they find that hematopoietic cells colonize the bone marrow after vessel invasion, and that LepR⁺ stromal cells –which are key component of the adult HSC niche- are absent from the fetal bone marrow. Using imaging analyses they propose that fetal HSC localize to arterioles. Analyses of their scRNAseq data suggest that AEC produce more HSC-supportive molecules than other types of endothelial cells. Selective ablation of Wntless –necessary for secretion of Wnt ligands- in AEC causes a ~20% reduction in functional HSC numbers. Conditional deletion of catenin b1

in hematopoietic cells leads to a ~4 fold reduction in phenotypical HSC and other progenitors. The authors also find that Wnt2 is the highest expressed Wnt ligand in AEC. Using a culture system they show that Wnt2 treatment promotes HSPC proliferation and colony formation. From these studies the authors conclude 1) that AEC are a niche for fetal HSC via Wnt ligand secretion and 2) “that fetal BM is fundamentally different from its adult counterpart with respect to existence of LepR⁺ niche cells, molecular interactions and the regulation of HSPCs”. The first statement is well supported by the experimental data –except cKIT/HSC imaging analyses that lack rigor. The second statement exaggerates the novelty of the authors’ findings and is not supported by the data.

Reply: We are very grateful for the summary of our main findings as well as insightful suggestions that have helped us to improve the manuscript.

Before our details point-to-point response to each question, we wish to clarify the issue regarding novelty of our work.

The aim of our scRNA-seq data is to provide an unbiased comparison of BM cells between E18.5 and adult. From our point of view, this will provide an important resource to understand the properties of fetal marrow and key features of BM development. As far as we know, this analysis has not been performed so far and is therefore novel.

In the original manuscript, we have focused mainly on bone marrow stromal cells (BMSC) and LepR⁺ cells because these populations show the most prominent differences between E18.5 and adult. We have now extended our analysis of endothelial cells (ECs) and hematopoietic stem/progenitor cells (HSPCs), which represent very important and major populations in marrow. We also show that the molecular interactions between stromal cells and HSPCs is substantially different at the interactome level for E18.5 or adult (Fig. 3e-3f). This also applies to Wnt signaling, where AEC-derived Wnt2 is specific for fetal BM (Fig. 5a-5b). From our point of view, these new analyses provide strong support for our conclusion “fetal BM is fundamentally different from its adult counterpart.

Previous work has indeed reported the absence of LepR⁺ cells by immunostaining and FACS. We cite and discuss the relevant references in the Discussion of the revised manuscript. However, our scRNA-seq analysis now shows that the absence of LepR⁺ cells not simply reflects the absence of a single or few molecular markers. Importantly, we find another population of vessel-associated mesenchymal cells in fetal BM, but these eBMSCs lack the expression of niche factors that are a hallmark of LepR⁺ cells. From our point of view, these findings are novel and will be very interesting for the research community.

Specific concerns and suggestions on how to improve the manuscript are detailed below:

Major (required)

Exaggerated claims of novelty: the central claim of the authors is that they demonstrate that the fetal microenvironment is fundamentally different from the adult microenvironment. This is based on the lack of LepR⁺ stromal cells in the fetal bone marrow and the finding that Wnt-producing AEC are a niche for fetal HSC.

1. The observation that LepR⁺ cells are not present in the fetal bone marrow has been documented in detail by various groups (see Dev Cell. 2014 May 12; 29(3): 340–349; Cell

Stem Cell. 2014 Aug 7; 15(2): 154–168; Nat Commun. 2019; 10: 3168). The authors should cite and discuss these manuscripts when presenting and interpreting their results.

Reply: Agree. We are very grateful for the reviewer to remind us of these important publications. We cite all these papers and have extended the discussion of previous studies in the revised manuscript.

As the reviewer mentions correctly, the absence of LepR⁺ cells in fetal bone marrow are indeed shown in (1) Figure 3A of Dev Cell. 2014 May 12; 29(3): 340–349, (2) Figure 2F of Cell Stem Cell. 2014 Aug 7; 15(2): 154–168, and (3) Figure 2a of Nat Commun. 2019; 10: 3168. While these were undoubtedly important findings, a couple of important questions were not addressed by these studies:

1. Are there other perivascular cells before the appearance of LepR⁺ cells?

In the field of vascular biology, it is well-established that recruitment of vessel-associated mesenchymal cells, mostly pericytes, is an integral part of angiogenic blood vessel growth in most organs (PMID: 21839917, 21925313). The vascular network in murine femur forms at E15.5 (Supplementary Fig.1a), but LepR⁺ cells become abundant only about 2 weeks after birth. It remained unknown whether other perivascular cells or BMSCs take the place of LepR⁺ cells. Another possibility was that reticular cells are present but lack LepR expression. Our scRNA-seq and immunostaining data indicate the existence of previously unknown BMSC sub populations (i.e. eBMSC-1 and eBMSC-2) in fetal BM. Notably, eBMSCs are the predominant BMSC population in fetal BM, representing about 66% of all fetal BMSC (Fig. 2a). The existence of these cell in fetal BM was not addressed by previous studies.

2. Can eBMSC replace niche function of LepR⁺ perivascular cells?

The discovery of eBMSC and absence of LepR⁺ cells in fetal BM raise the question whether eBMSC transiently take over the important role of the LepR⁺ population as niche cells for HSPCs. Based on our scRNA-seq data, eBMSC have rather limited expression of major niche factors including SCF, Cxcl12, Angiopoietin-1 and interleukin-7. Moreover, eBMSC show approximate 2000 DEGs distinguishing them from adult LepR⁺ BMSCs.

3. Does the eBMSC population continue to exist in adult BM?

Another question is whether eBMSC continue to exist in adult BM. Our scRNA-seq analysis suggest that this is the case even though their percentage drops sharply from 66% (E18.5) to 2.7% (adult). There is also a substantial number of DEGs (419) between fetal and adult eBMSC-1 cells.

4. Are other major BM stromal cell populations absent in the fetus?

Although previous literature recorded the absence of LepR⁺ cells in fetal BM, the differences between fetal and adult BM remain little understood. Based on our scRNA-seq analysis, we propose that all other major BM cell populations (subpopulations) are already present at E18.5, which also means that the absence of LepR⁺ reticular cells is rather unique and peculiar.

Taken together, our manuscript provides insight into many important and previously unknown aspects of bone marrow development.

2. In the adult LepR⁺ osteoblastic stromal cells, sinusoidal endothelial cells, and arteriolar endothelial cells cooperate to maintain HSC. Coskun et al., -using *Osx* knockout mice- demonstrated that osteoblastic stromal cells are required for HSC colonization of the fetal liver (reference 19). The authors' discovery that -similar to adult AEC (Nat Commun. 2018 Jun 22;9(1):2449) – fetal AEC maintain fetal HSC suggest that the fetal BM microenvironment is actually similar to the adult microenvironment. Note that the Frenette lab showed that arterioles selectively maintain HSC in the fetal liver [Science. 2016 Jan 8; 351(6269): 176–180]. The aorta is the site of emergence of HSC in the embryo, and arterioles support both fetal liver and adult HSC. The discovery that arterioles also support fetal BM HSC –while important- seems incremental. These studies should also be considered by the authors when discussing their results.

Reply: We agree that there are a couple of landmark studies, which have shown that arteries are relevant for certain aspects of the development and function of the hematopoietic system. Some of this work has involved genetic tools that were developed and shared by us (e.g. Nat Commun. 2018 Jun 22;9(1):2449, Reference 36).

We respectfully disagree that our findings are incremental because our work addresses a completely different biological process – the first colonization of BM by hematopoietic cells – and identifies a new artery-derived signal. In the revised manuscript, we discuss previous findings and the novelty of the current study.

We also show important differences in arterial gene expression between fetal and adult samples (Fig. 4g and Supplementary Fig. 6a). Among other factors, *Wnt2* is provided by AECs at E18.5 but expression is substantially reduced and therefore potentially irrelevant in the adult (Fig. 5a). Preliminary data shows that there are no major haematopoietic defects in adult *Wls*^{iΔBmx} mutants (Reviewer 2 Figure 1, see next question).

We are aware of the important role of arteries during endothelial-hematopoietic transition in the aorta-gonad-mesonephros region, fetal liver development and adult HSC regulation. While the regulation of adult HSPC is investigated extensively, the mechanisms mediating HSCs engraftment and expansion in the initial stages of BM development remain little understood. Our data provide direct insights into the interaction between arteries and HSCs in fetal BM, which are important and new.

The references mentioned by the reviewer are cited in revised manuscript and mentioned in the Discussion.

It is possible –as the authors suggest- that there is a differential requirement for AEC-derived Wnt ligands between fetal and adult HSC. If the authors wish to make this claim then they should demonstrate that Wntless deletion in adult AEC does not perturb adult HSC.

Reply: Agree. We follow reviewer's suggestion and analyzed young adult *Wls*^{iΔBmx} mice (at about 8 weeks of age). We detect a mild but statistically significant reduction of HSCs and LSK cells in mutants relative to littermate controls (Reviewer 2 Figure 1). However, we did not detect significant changes in mature or progenitor hematopoietic cells (Reviewer 2 Figure

1). This evidence suggests that artery-derived Wnt signal is less relevant in the adult organism, which may well reflect the influence of other signals derived from ECs and LepR+ reticular cells.

In our scRNA-seq analysis of Wnt expression, we detect large differences between E18.5 and adult AECs (Fig. 5a). Notably, *Wnt2* is no longer the most prominent Wnt ligand in adult and there are also changes in the interactome analysis (Fig. 5a and 5b).

Overall, these results indicate difference in the role of arteries in the fetal relative to the adult organisms. The role of AEC-derived Wnt in adult steady state hematopoiesis but also in the challenged BM is indeed an interesting question, which is, however, reaching beyond the scope of the current study.

Reviewer 2 Figure 1 Analysis of hematopoiesis in young adult *Wls*^{iΔBmx} mice

Biological significance: The authors elegantly demonstrate that fetal AEC maintain HSC. However, the observed reduction in HSC numbers and function is modest (Fig. 3c-e). Does this reduction translate into an overall defect in fetal or adult hematopoiesis? Are mature blood cells reduced in the marrow/blood of fetal or adult *Wls*^{iΔBmx} mice? In other words: does perturbation of the fetal arteriolar niche result in fetal or adult hematopoietic injury? This would greatly increase the significance of the authors' findings.

Reply: First of all, we would like to point out that the *Bmx-CreERT2* line targets the endothelium of larger and more mature arteries, whereas the smallest arterioles are spared. In addition, there might be other, non-arterial sources of Wnt in fetal BM, which might explain why reduction in HSC number is modest relative to other mutants.

To check whether the HSC reduction resulted in defective fetal hematopoiesis, we examined the mature hematopoietic cells in bone marrow and peripheral blood of *Wls*^{iΔBmx} embryos. In fetal BM, we detect a significant reduction of Ter119% erythrocytes together with increase of CD45%, Gr-1% and B220%, suggesting that the balance of erythrocytes and leukocyte generation is disrupted in *Wls*^{iΔBmx} mutants (Fig. 5j). In contrast, no significant changes to CD45%, Gr-1%, B220% and CD4+CD8% were seen in *Wls*^{iΔBmx} peripheral blood (Supplementary Fig. 8b). This may also reflect that there is no apparent phenotype in

Wls^{iΔBmx} fetal liver, which plays an important role in hematopoiesis at this developmental stage.

Concerns regarding imaging analyses:

The authors extensively use imaging of c-Kit⁺ cells to examine HSPC colonization of the bone marrow as well as to quantify reductions in HSPC numbers in their mouse models. However –at least to this reviewer- the c-Kit staining seems inconsistent/not specific. For example, in multiple panels (e.g. Fig. 1c; Fig. 3b,f, I; Extended Data Fig. 2e-g; Extended Data Fig. 7a) c-Kit⁺ cells are uniformly distributed through the bone marrow, essentially contiguous to each other. In sharp contrast, the flow cytometry data presented by the authors (Extended Data Fig. 2j) shows that c-kit⁺ HSPC are rare. This FACS frequency is consistent with previous studies by the Hirschi (reference 19) and Weissman [PLoS Biol. 2004 Mar;2(3):E7] groups. The discrepancy between the authors FACS and imaging data indicates that either a) the CD117 imaging is not specific; or b) that there are numerous c-Kit⁺ cells present in the bone marrow that are not HSPC (or even hematopoietic cells). In this case the data

shown in panels Fig. 1c; Fig. 3b,f, I; Extended Data Fig. 2e-g; and Extended Data Fig. 7a will need to be extensively corrected. Of note when the authors fate map c-Kit⁺ cells using Kit-CreER mice they find that cKit⁺ cells are rare (Extended Data Fig. 9e) further suggesting that the cKit antibody stain is not specific. I strongly encourage the authors to extensively validate the specificity of all their hematopoietic cell stains as the data presented is not convincing.

Reply: We fully understand the concern of the reviewer and have taken great care to validate the antibody.

1. This antibody (Biolegend, Catalog number 105806; <https://www.biolegend.com/en-us/products/fitc-anti-mouse-cd117-c-kit-antibody-74?GroupID=BLG1945>) is FITC-conjugated. We have not used any secondary antibody to enhance the FITC signal, which eliminates one common source of unspecific staining. This antibody is generated from the “2B8” clone, which has been tested and used in numerous peer-reviewed publications (please see the link of manufacturer above).

2. We fully agree with the reviewer that c-Kit⁺ cells are scarce in fetal BM. The reason for what appears as an “inconsistency” is that we show projected low magnification images of thick sections (120μm) of fetal bone marrow in Fig. 1a-1c, Fig. 5d, Fig. 6a, Fig. 6e, Supplementary Fig. 2a-2g, Supplementary Fig. 9a, and Supplementary Fig. 10p. In the past, we have started to work with thick tissue sections to capture 3D information and thereby gained a lot of insight into the architecture of the bone vasculature. The advantage of this approach is that (1) images show the overall number and distribution of c-Kit⁺ cells in BM and (2) that the image is objective without subjective selection of a small area in BM. In the original data, c-Kit⁺ cells are found in different spatial layers, but they appear to be close to each other in maximum intensity projections. Similar images are also shown in previous publications investigating adult BM where c-Kit/CD117⁺ cells appear more abundant than in thin sections or based on FACS data (Reviewer 2 Figure 2).

Nevertheless, we performed an extensive validation of the antibody by FACS and immunostaining to confirm that the signal is from CD45⁺ hematopoietic cells but not stromal cells (see below).

3. To validate the antibody, we used an APC-conjugated c-Kit antibody from another provider (BD, cat. no. 553356) for FACS analysis. c-Kit FITC and c-Kit APC antibodies label the same population of cells (Reviewer 2 Figure 3a), whereas we did not detect a Kit-FITC⁺ Kit-APC⁻ population.

4. In the immunostainings, c-Kit signals show co-localization with CD45 signal, indicating the labeling of hematopoietic cells (arrows in Reviewer 2 Figure 3b). Likewise, CD150 signals co-localize with c-Kit staining (arrows in Reviewer 2 Figure 3c), which also indicates that the c-Kit antibody labels predominantly hematopoietic cells. Furthermore, the c-Kit signal does not co-localize with CD31⁺ endothelial cells, PDGFR β ⁺ mesenchymal cells and Osterix⁺ osteoprogenitor cells (arrows in Reviewer 2 Figure 3h-3j). In addition, c-Kit signals are largely excluded from mature hematopoietic cells including B220⁺, Ter119⁺ and CD3e⁺ cells (arrows in Reviewer 2 Figure 3e-3g).

5. c-Kit FITC signal is predominantly seen at the cell surface (Supplementary Fig. 2h), which is characteristic for receptor tyrosine kinases.

6. In thinner sections (30 μ m) and at higher magnification, it can be easily seen that the number of c-Kit⁺ cells is significantly lower than Ter119⁺ red blood cells and CD45⁺ hematopoietic cells in fetal BM (Reviewer 2 Figure 3b and 3e). The image for Kit-CreERT2 mice is also obtained from thinner sections. From our point of view, the number of cells labeled by GFP and c-Kit antibody is not substantially different (comparing Supplementary Fig. 11d with Reviewer 2 Figure 3b and 3d-3e). We have added the statement “120 μ M thick sections were used to acquire overview images showing the distribution of c-Kit⁺ cells in fetal BM” to the section on “Cryosections, immunohistochemistry, confocal imaging and 3D-reconstruction” of the Methods.

Reviewer 2 Figure 3 Validation of c-Kit antibody

7. At E16.5, the c-Kit signal is very scarce in bone marrow (Fig.1a and Fig.4d). However, endothelial cells, bone marrow stromal cells and osteoprogenitor cells are already present at this stage (Reviewer 2 Figure 3h-3j). This further supports that the c-Kit antibody is not labeling these non-hematopoietic cells in fetal BM.

8. To quantify the reduction of HSPCs in our mouse model, we have not merely relied on immunostaining with c-Kit antibody. In fact, FACS has been used to quantify the fold change of HSPCs in *Wls^{iΔBmx}* mice (Fig. 5g and 5i), *Ctnnb1^{ΔVav1}* mice (Fig. 6c and 6d) and *Ctnnb1^{iΔKit}* mice (Fig. 6g and 6h). The reduction of HSCs in *Wls^{iΔBmx}* mice is also validated by CD45.2/CD45.1 competitive repopulation assay (Fig. 5e-5f).

9. Other labs have used the same cKit-FITC antibody for immunostaining of c-Kit (Figure 5b in PMID: 30444930, or see Reviewer 2 Figure 4). We can clearly see co-localization of CD117 signal with another hematopoietic cell marker. Similar to our own results, the CD117 signal is predominantly seen at the cell surface.

Reviewer 2 Figure 4 Immunostaining of the same c-Kit antibody in other publication (PMID: 30444930)

Taken together, we have extensively validated the specificity of the antibody by FACS and immunostaining. This antibody labels HSPCs but not ECs, stromal cells and osteoprogenitor cells in fetal BM. The same antibody has been used in published work by other groups for FACS and immunostaining. What might be perceived as inconsistency between FACS and immunostaining results, reflects that we have used thick sections for the latter.

In Fig. 2d-g the authors image Lin-CD48-CD41-CD150+ HSC or Kit+ cells with arterioles at E16.5. They also measure the distance from each HSC or DAPI+ to the closest artery and compare the observed distributions. From this experiment the authors conclude that most HSC localize near arterioles. These analyses are very superficial. To the best of this reviewer knowledge this is the first time that fetal HSC are imaged in the bone marrow. This is a monumental achievement that is barely examined in the manuscript. At this developmental time only 3-5 HSC are expected to be present in the bone marrow. What is HSPC localization with other components of the microenvironment (e.g. sinusoids, or stromal cells)? Are HSC evenly distributed through the BM? Are HSC relocating away from arterioles after Wls deletion? Although the authors claim to have imaged over 50 HSC only one image is shown (Fig. 2d). More images will help the reader to better understand HSC distribution in the fetal bone marrow. Comparing the distances of HSC with DAPI+ cells is incorrect as –at this developmental time point- many DAPI+ cells are stromal or endothelial cells (as shown by the authors). HSC distances should be compared with those of randomly placed cells using the Lin/CD48/CD41+ or cKit+ signal to define the possible locations of the random cells, statistical comparisons should be provided (see Nature 526, 126–130 (2015)).

Reply: For better visualization of HSC location in fetal BM and similar to previous publications (PMID: 26416744), we have used Imaris for 3D reconstructions and quantification in revised manuscript (Supplementary Fig. 5c-5g and Methods). Examples of original images and 3D-reconstructed models are displayed in Supplementary Fig. 5c.

At E16.5, HSCs as well as HSPCs are mainly located in the bone marrow of the central diaphysis and not in the metaphysis and endosteum (Supplementary Fig. 5f). This is different from HSC localization in the adult, which includes the BM cavity, endosteum as well as transition area between metaphysis and diaphysis (PMID: 32025033, 26416744).

At E16.5, HSCs and c-Kit+ HSPCs are located close to CD31+ BM vessels and PDGFRβ+ perivascular stromal cells (Supplementary Fig. 5f). More specifically, HSCs and c-Kit+ HSPCs are found very close to Caveolin-1+ arterial endothelial cells (Fig. 4a-4f). Statistically, about 70% of HSCs and 80% of c-Kit+ HSPCs are found within a distance of 10µm to Caveolin-1+ arteries. In contrast, many Caveolin-1 negative blood vessels do not

show an association with HSCs (Fig. 4a and 4d). Likewise, HSCs are not close to Osterix+ cells at E16.5 (Supplementary Fig. 5f). At E18.5, however, a small fraction of HSCs locates closer to Osterix+ osteoprogenitor cells indicating dynamic changes in developing BM (Supplementary Fig. 5g).

Regarding the mutant analysis, the distance of HSCs to Caveolin-1+ arteries is not statistically different between *Wls^{iΔBmx}* mutants and littermate controls at E16.5, (Supplementary Fig. 7j). This is different at E18.5 when the distance is significantly increased in *Wls^{iΔBmx}* BM (Supplementary Fig. 7k and Fig. 5h). Thus, AEC-derived Wnt contributes to the positioning of HSCs during fetal BM development.

We agree with the reviewer that it is a good idea to use mature hematopoietic cells (Lin+ CD48+ CD41+) as a control for HSC distance comparisons. However, as the location of mature hematopoietic cells in fetal BM is not completely random (please see Supplementary Fig. 5d), we decided to keep DAPI as another control. In both cases, the distance between HSCs and Caveolin-1+ arteries is significantly shorter than to DAPI-labelled or mature hematopoietic cells (Fig. 4c). The same applies to c-Kit+ HSPCs (Fig. 4f). Statistical analysis was done with the Kolmogorov-Smirnov test for all distance comparisons and a *p*-value less than 0.05 is considered to be statistically significant (Fig. 4c, 4f, 5h, Supplementary Fig. 7j).

Bioinformatic analyses: The single-cell analyses were performed using well-established and rigorous methods, without evidence of clear batch effects or over-clustering, with results reasonably validated. However, the analyses presented are superficial and missing in depth.

The authors claim that “In contrast to the observed differences in BMSC subpopulations, major HSPC subsets are detected at only slightly different ratios both in embryonic and adult bone samples (Extended Data Fig. 5a-5d). At the same time, DEG analysis indicates relatively minor alterations at the transcriptional level and preserved hierarchical organization of embryonic and adult HSPCs (Extended Data Fig. 5e-5g)”. This is a major claim that directly contradicts recent studies from the Magee lab and Tan labs showing profound differences in the transcriptional program of fetal and adult HSC [Cell Stem Cell. 2020 Nov 5;27(5):732-747; Cell Rep 2019 Dec 17;29(12):4200-4211.e7] and numerous previous studies showing functional and transcriptional differences between fetal and adult HSC (see Exp Mol Med. 2013 Nov 15;45(11):e55). This needs to be addressed in detail.

Reply: We apologize in case our previous description has caused any confusion. We are not trying to claim there are no differences between fetal and adult HSCs. In fact, there are substantial differences in gene expression and this part of the analysis has now been expanded (see Fig. 1f-1h and our response to question 19 by reviewer #1). While there are also some interesting differences in the relative representation of certain HSPC subsets, the major subpopulations and HSPC hierarchy are already established in fetal BM (Fig. 1f).

The revised manuscript now contains DEG analysis of BMSCs, HSPCs and ECs at the sub-cluster level (Fig. 1h, Fig. 2f, Fig. 3c). The comparison adult and fetal HSPC sub-clusters reveals thousands of DEGs, which include important transcription factors (e.g. GATA family, Lmo2, ETS family) and receptors (e.g. thrombopoietin receptor, CSF-1 receptor) known to regulate hematopoiesis and HSPC development (PMID: 33193725, 33103827). Taken together, our findings regarding fetal HSPCs are consistent with the existing literature. We

did not focus much on the characterization of fetal HSPC subsets because this is not the main focus of the current manuscript.

Accordingly, we have our description of the relevant finding: “*Although major HSPC subsets are present at E18.5 (Fig. 1f-1g; Supplementary Fig. 3f-3g), myeloid and megakaryocyte progenitors are less abundant than in adult BM. Moreover, differentially-expressed gene (DEG) analysis at subcluster level shows substantial differences between E18.5 and adult HSPCs with more than 500 to 2000 DEGs in each subcluster (Fig. 1h), which is consistent with previous reports^{23,27,28}.*”

Similarly the classification of the different hematopoietic populations seems superficial; why are erythroid progenitors not detected in the adult of fetal bone marrow? Can the authors provide more detailed clustering information? Are the scRNAseq subsets substantiated by FACS?

Reply: In both E18.5 and adult bone marrow, mature hematopoietic cells are the major population, which prevents direct scRNA-seq analysis of all cells. In our hands, depletion of mature hematopoietic is indispensable for BM stromal cell analysis and similar approaches have been used in previous publications (PMID: 31130381, 31871321). In this study, we depleted Lin⁺ cells with the help of antibody-conjugated magnetic beads or FACS sorting prior to scRNA-seq analysis (Supplementary Fig. 3a).

After depletion of most mature hematopoietic cells, the remaining erythrocytes or erythroid progenitor cells are detected in both E18.5 and adult samples (Fig. 1d). Markers include the expression of hemoglobin family genes (Hbb, Hba) and carbonic anhydrase family genes (Car1, Car2) (Supplementary Fig. 3b-3c). However, these cells are substantially different from other HSPCs and were classified as a different cluster in the initial unsupervised clustering.

Detailed information regarding mature hematopoietic cells depletion and clustering information are provided in Methods. The classification of cells relies on unbiased and unsupervised methods. The same significant principal components used for UMAP non-linear dimensional reduction were also used for the unsupervised hierarchical clustering analysis by FindClusters function in Seurat package. We tested different resolutions between 0.1 ~ 0.9 and selected the final resolution using clustree R package. Cellular identity for each cluster was determined by finding cluster-specific marker genes in FindAllMarkers function with minimum fraction of cells expressing the gene over 25% (min.pct=0.25).

Classification results were validated by marker expression. For example, the higher expression of corresponding marker for myeloid progenitor (Prss34^{high}), megakaryocyte progenitor (Pf4^{high}), multipotent progenitor (Kit^{high}, Hlf^{high}, CD34^{high}) and neutrophil progenitor (S100a8^{high}) can be seen in feature plots (Supplementary Fig. 3g). More markers are shown in Supplementary Fig. 3f. These markers were previously validated and used in various scRNA-seq publications (e.g. PMID: 31871321, 29915358, 26627738, 29588278).

The authors show expression of HSC-supportive factors in endothelial cells. Why is this information not provided for the rest of the stromal analyses? This will be very helpful for the field as other stromal cells are expected to maintain HSC via these signals [for example the

authors' own data shows that other cells produce Wnt ligands that maintain HSC and the phenotype of the hematopoietic *Ctnfb1* cKO is much stronger than the AEC *Wlsc*KO (Fig. 3g-k)]

Reply: Agree. In the revised manuscript, we have added UMAP feature plots showing HSC-supporting factor expression pattern in all E18.5 cells (Supplementary Fig. 6b).

While our study emphasizes the role of AECs during early HSPC engraftment and expansion in fetal BM, this does not rule out important functions of other stromal cell populations through the release of Wnt ligands or other factors.

We have therefore added the following statement to the Discussion: *“In addition to AECs, other stromal cell populations are likely to contribute to the haematopoietic colonization of fetal BM, which may involve the release of Wnt ligands as well as other molecular signals. This might also explain why the phenotypes seen in *Ctnnb1*^{ΔVav1} or *Ctnnb1*^{ΔKit} mutants are more severe than the haematopoietic defects in *Wls*^{ΔBmx} embryos.”*

Minor:

- The cell-to-cluster assignments, monocle pseudo-time predictions and cluster-marker genes should be provided in supplemental Excel files to enable reproducibility.

Reply: Meta information including cell to cluster and sample assignments of the data and cluster marker gene lists for each analysis are provided as supplementary tables.

Individual cell-to-cluster assignment is provided in Supplementary Table S3. Cluster-marker genes are provided in Supplementary Table S4.

Monocle pseudo-time predictions is removed in the revised manuscript.

- More detailed single-cell metrics for each sample should be provided along with the number of cell barcodes/total reads/UMI's detected or targeted for sequencing using the BD Rhapsody system. Methods for the targeted sequencing are not sufficiently detailed.

Reply: Thank for bringing this to our attention. We have updated the Methods accordingly. Additional information of sequencing metrics for all samples of whole transcriptome (embryo and adult) and targeted analysis (embryo BMSCs, embryo ECs and adult BMSCs) including number of cell barcodes/total reads/UMI's detected is provided in Supplementary Table 5.

- The rationale for targeted sequencing is not clear. Does this provide better resolution for WNT signaling component genes? UMI metrics should clarify.

Reply: Because many of the relevant genes exhibit relatively low abundant transcripts, 3' whole transcriptome analysis is not sensitive enough to detect expression of all ligands and other pathway components. We therefore utilized a custom panel for targeted scRNA-seq because this approach makes use of nested gene-specific primers and is therefore by far more

sensitive. A FASTA file of the targeted panel is provided as supplementary table (Supplementary Table 6). Further technical details and information regarding data processing is provided in the section “Single cell RNA-seq data analysis” and “Targeted single-cell RNA-seq analysis of Wnt pathway genes” in the Methods. Sequencing metrics for the targeted analysis is provided in Supplementary Table 5, as mentioned above.

- In Figure 4b, how were cell populations determined? Were these aligned to the whole transcriptome data for cell assignment using a reasonable set of markers for delineating?

Reply: We included primers for cell type-specific markers in the targeted custom panel (Supplementary Fig. 7a and Supplementary Table S6).

- In many panels of Figs 1-2 the authors use the term “sinusoidal” instead of “sinusoidial”.

Reply: Thank you very much for alerting us to this mistake, which has been corrected.

- To better substantiate that the *Wls^{ΔBmx}* phenotype is specific to the BM and not initiated in the fetal liver the authors should perform competitive HSC transplants from *Wls^{ΔBmx}* and control fetal liver.

Reply: Agree. Following the reviewer’s suggestion, we have used *Wls^{ΔBmx}* and littermate control fetal liver of as CD45.2 donor mixed with CD45.1 donor cells for transplantation into lethally-irradiated CD45.1 mice (Supplementary Fig. 8e). At 4 months after transplantation, chimerism of myeloid cells, B cells and T cells was not significantly different between mutant and control (Supplementary Fig. 8e-8f).

- Throughout the manuscript the authors normalize HSC and hematopoietic cell numbers to that of control mice. The reason for this is unclear and complicates interpretation of the data. Please show absolute numbers per bone for each population analyzed.

Reply: The reason for normalizing the cell number is that HSPCs are rapidly proliferating at E18.5. In adult mice, the number of HSCs and HSPCs is relatively stable, whereas the rapid increase of these cells between E16.5 to E18.5 (Fig.1a-1c and Supplementary Fig. 2a-2e) results in substantial variation in HSPC number from litter to litter depending on the exact time of fertilization. This is an intrinsic issue that cannot be circumvented even though breeding pairs were immediately separated after detection of a vaginal plug and staging of embryos based on anatomical criteria was used.

Please see below our quantification of Lin- Kit⁺ HSPC cell number as an example for the quantification with and without normalization (Reviewer 2 Figure 5). In all 5 different pregnant females, the average number of Lin- Kit⁺ cells is lower in *Wls^{ΔBmx}* embryos compared to littermate controls from the same pregnant female. However, the *t*-test result for

absolute cell number is 0.16 because of the big variation between different litters. The issue of litter-to-litter variation is resolved by normalization.

Pregnant Female	Bmx+;/; Wls f/f	Lin-Kit+ number	Lin-Kit+ number	Normalized Lin-Kit+ number	Bmx+;/; Wls f/f	Lin-Kit+ number	Lin-Kit+ number	Normalized Lin-Kit+ number
1#	1#	182	182	0.994535519	1#	90	90	0.491803279
	2#	184	184	1.005464481		average	90	
	average	183				average	90	
2#	1#	202	202	0.990196078	1#	88	88	0.431372549
	2#	206	206	1.009803922	2#	132	132	0.647058824
					3#	204	204	1
	average	204			average	141.3333333		
3#	1#	568	568	1	1#	350	350	0.616197183
					2#	512	512	0.901408451
	average	568			average	431		
4#	1#	454	454	1.407024793	1#	166	166	0.51446281
	2#	168	168	0.520661157	2#	260	260	0.805785124
	3#	346	346	1.07231405				
	average	322.6666667			average	213		
5#	1#	304	304	1.106796117	1#	194	194	0.70631068
	2#	226	226	0.822815534	2#	242	242	0.881067961
	3#	294	294	1.07038835	3#	84	84	0.305825243
					4#	174	174	0.633495146
	average	274.6666667			5#	262	262	0.953883495
				average	191.2			
		Ttest	0.164575004	0.001538814				

Reviewer 2 Figure 5 Quantification of Lin- Kit+ cells

- The authors frequently refer to their samples as “embryonic” even though analyses are performed at E16.5-18.5 (late fetal period).

Reply: To our knowledge, the stage-dependent discrimination between embryo and fetus applies to human development and is not used in the same stringent fashion for mice. For example, in Theiler’s "The House Mouse: Atlas of Mouse Development" (Springer, New York, 1989) or in “Mouse Development: Patterning, Morphogenesis, and Organogenesis” by Rossant and Tam (Academic Press 2002), which are seminal textbooks on mouse development, the word “embryo” is used for all stages.

Reviewers' Comments:

Reviewer #1:

Remarks to the Author:

The revised version of the manuscript by Liu, Cheng and Jeong has notably improved. The creation of shorter sections makes the manuscript much easier to read and clearer. It is noticeable that there is a good effort by the authors to address the reviewers comments.

My initial comments are mostly satisfied. However, there are a few questions remaining.

1. Overall, the comparison of transcriptional profiles is very superficial. There is no interpretation of the results from these comparisons and as it stands, they do not add much valuable information as the analysis is limited to indicate the amount of DEGs. When an interpretation is attempted, it is minimal and vague. For example, in the newly added paragraph in the section "Comparative single cell analysis of HSPCs and stromal microenvironments" it is necessary to be more specific about the clusters where the transcription factors are differentially expressed (see comment for Fig. 1i). What are the consequences of these differences?

2. In this version of the manuscript, DEGs are defined with $|\log_2FC| > 0.5$ (1.4 fold change) which is a very permissive cut off for this kind of data. I would suggest using at least > 1 . Also, in order to get robust and reproducible results in this type of dataset, it is very recommendable to include an additional threshold related to the mean expression of the genes. I would strongly suggest plotting MA plots (mean expression vs FC) to define a cut off to remove the lowly detected/expressed genes that are more prone to dropouts and therefore to more variation. This threshold should be indicated in the appropriate heading within the methods section.

3. The authors state that Wnt2 improves transplantation efficiency. See question 39 in the rebuttal letter. The results show that, even though Wnt2 has an effect on HSPC proliferation and colony formation, it does not improve transplantation efficiency since there are not differences when equal number of cells are transplanted. In line with this, it is indicated by the authors in their rebuttal letter: "treatment with recombinant Wnt2 induces an increase in HSPC number, whereas we have no evidence that it enhances the stem cell properties and repopulation potential of individual cells". Therefore, this statement should be removed or altered to reflect the actual findings of the experiments.

4. Fig. 1g and 2a.- Include statistical analysis for the comparison in composition between clusters.

5. Fig. 1i and 2g.- Include statistics or highlight in which clusters the genes are differentially expressed.

6. Supplementary Figure 3i.- I cannot see the benefit of including the comparison using Metacell as it is at the moment. It does not add any information, and when looking more carefully to the figure some discordances appear. In particular, megakaryocytes are completely disconnected from multipotent HSPCs which wouldn't agree with the current knowledge in the field. The UMAP is actually much more informative. Clearly, fetal HSPCs lack a group of megakaryocyte progenitors (the authors describe them as Cd41+Itga6+ in question 19 of the rebuttal letter), which it is certainly biasing the comparison for this cluster in figure 1h.

7. No mention of Suppl Fig 4h in the text.

8. Fig 3e and 3f.- No interpretation in the text.

9. It is stated "more than 75% of the HSC population is found within a distance of 10 μ m from Caveolin-1+ AECs (Fig. 4c; Supplementary Fig. 5d),". The revised version of figure 4c indicates less

than 70%. Please, amend.

Reviewer #2:

Remarks to the Author:

The authors have addressed all of my concerns. Congratulations on an excellent manuscript.

REVIEWERS' COMMENTS

Reviewer #1 (Remarks to the Author):

The revised version of the manuscript by Liu, Cheng and Jeong has notably improved. The creation of shorter sections makes the manuscript much easier to read and clearer. It is noticeable that there is a good effort by the authors to address the reviewers comments.

My initial comments are mostly satisfied. However, there are a few questions remaining.

Reply: Thank you very much for this positive assessment of our revision and the additional suggestions. Point-to-point responses to the remaining questions are provided below.

1. Overall, the comparison of transcriptional profiles is very superficial. There is no interpretation of the results from these comparisons and as it stands, they do not add much valuable information as the analysis is limited to indicate the amount of DEGs. When an interpretation is attempted, it is minimal and vague. For example, in the newly added paragraph in the section “Comparative single cell analysis of HSPCs and stromal microenvironments” it is necessary to be more specific about the clusters where the transcription factors are differentially expressed (see comment for Fig. 1i). What are the consequences of these differences?

Reply: Thank you very much for this comment. We appreciate that our analysis of HSPCs could be more extensive, but this manuscript focuses on the mechanism regulating the engraftment of HSCs and HSPCs into fetal BM (Figure 4-7). This mechanism and phenotype rely, at least in part, on the differences between the adult and fetal BM microenvironment (Figure 1-3), which most prominently concern the absence of LepR⁺ reticular cells (Figure 2) and special features of arteries in the fetus (Figure 4g, 5a-5b). We provide extensive experimental evidence for these differences to provide novel and conceptual insight into a previously poorly characterized developmental process.

In contrast, the transcriptomic differences between fetal HSPCs and adult HSPCs are already addressed by multiple existing publications focusing on the properties of fetal hematopoietic stem and progenitor cells (e.g. Li et al., Cell Stem Cell 2020, PMID: 32822583; Ranzoni et al., Cell Stem Cell 2021, PMID: 33352111; Popescu et al., Nature 2019, PMID: 31597962). At the same time, there is also increasing evidence that approaches based solely on scRNA-seq have substantial limitations. For example, it has been shown that scRNA-seq alone fails to fully resolve subtle changes involved in early hematopoietic differentiation (Weinreb et al., Science 2020, PMID: 31974159) so that other approaches need to be combined with this method. Lineage trajectories are computationally inferred and do not necessarily reflect clonal relationships between cells (Weinreb et al., Science 2020, PMID: 31974159).

The reasons above should explain why we have focused on the role of stromal cells in fetal marrow. As suggested by the expert reviewer, we have already added some analysis of fetal HSPCs during the first revision, but, due to the reasons mentioned above, there is little justification for further expansion of this part.

2. In this version of the manuscript, DEGs are defined with $|\log_2FC| > 0.5$ (1.4 fold change) which is a very permissive cut off for this kind of data. I would suggest using at least > 1 . Also, in order to get robust and reproducible results in this type of dataset, it is very recommendable to include an additional threshold related to the mean expression of the genes. I would strongly suggest plotting MA plots (mean expression vs FC) to define a cut off to remove the lowly detected/expressed genes that are more prone to dropouts and therefore to more variation. This threshold should be indicated in the appropriate heading within the methods section.

Reply: Agree. As suggested by the reviewer, we now use $|\log_2FC| > 1$ as threshold. We also use p-adjusted value=0.001 to replace p=0.05 as another threshold. Therefore, the threshold is much more stringent to get robust and reproducible results. In the more stringent threshold, we could still detect hundreds of DEGs in most of sub-cluster comparison (Fig. 1h, 2f and 3c) We did also follow the reviewer's suggestion to use MA plots (mean expression vs FC) to display our results. The threshold is directly indicated in figures to facilitate reading.

3. The authors state that Wnt2 improves transplantation efficiency. See question 39 in the rebuttal letter. The results show that, even though Wnt2 has an effect on HSPC proliferation and colony formation, it does not improve transplantation efficiency since there are not differences when equal number of cells are transplanted. In line with this, it is indicated by the authors in their rebuttal letter: "treatment with recombinant Wnt2 induces an increase in HSPC number, whereas we have no evidence that it enhances the stem cell properties and repopulation potential of individual cells". Therefore, this statement should be removed or altered to reflect the actual findings of the experiments.

Reply: I think that we fully agree about the meaning of the data. Wnt2 pretreatment of cells *ex vivo* leads to the expansion of HSPCs and thereby accelerated BM colonization after transplantation. In other words, if one starts with a certain number of cells at the beginning of the experiment, the Wnt2 treatment group performs better after transplantation.

This is also consistent with the Wnt2 treatment of FACS-isolated LSK cells (Fig. 7a-7b) or cultured adult or neonatal LSK cells (Fig. 7c-7d; Supplementary Fig. 13a-13b), which lead to increases in the number of HSCs and LSK cells.

We have modified the wording of the sentence and removed the word "efficiency" to avoid any misunderstanding.

4. Fig. 1g and 2a.- Include statistical analysis for the comparison in composition between clusters.

Reply: In our analysis, we pooled different biological samples in each group (E18.5=8; adult=4) to obtain sufficient amounts of stromal cells and minimize variation caused by individual samples. As these scRNA-seq experiments did not involve further independent biological replicates, statistical comparison for the composition between clusters is not feasible.

5. Fig. 1i and 2g.- Include statistics or highlight in which clusters the genes are differentially

expressed.

Reply: Clusters are actually shown on the x-axis in both figures so that this information is already provided. The *p-values* for each subcluster have been added to the figures (Fig. 1i and Fig. 2g) in the revised version of manuscript.

6. Supplementary Figure 3i.- I cannot see the benefit of including the comparison using Metacell as it is at the moment. It does not add any information, and when looking more carefully to the figure some discordances appear. In particular, megakaryocytes are completely disconnected from multipotent HSPCs which wouldn't agree with the current knowledge in the field. The UMAP is actually much more informative. Clearly, fetal HSPCs lack a group of megakaryocyte progenitors (the authors describe them as Cd41+Itga6+ in question 19 of the rebuttal letter), which it is certainly biasing the comparison for this cluster in figure 1h.

Reply: We agree with the reviewer. Suppl. Fig. 3i has been removed in the revised version of manuscript.

As mentioned above, computationally inferred lineage trajectories do not necessarily reflect clonal relationships between cells (Weinreb et al., Science 2020, PMID: 31974159) and additional experiments, such as genetic lineage tracing, are required for meaningful results. Our data simply indicates that gene expression of the sequenced megakaryocyte progenitors is rather distinct from the other cell populations. As the lineage relationships of cells in the hematopoietic system are well established by a huge body of literature and a diverse range of experimental approaches, we do not see sufficient justification for extending this part of the work.

7. No mention of Suppl Fig 4h in the text.

Reply: Thank you for alerting us to this oversight, which has been corrected. We have also modified this panel in the figure to put more emphasis on the differences between fetal and adult BMSCs. Suppl. Fig. 4h is now mentioned on page 6 of the main text.

8. Fig 3e and 3f.- No interpretation in the text.

Reply: We included additional statements for Fig. 3e and 3f on page 7 of the main text.

9. It is stated "more than 75% of the HSC population is found within a distance of 10 μ m from Caveolin-1+ AECs (Fig. 4c; Supplementary Fig. 5d)". The revised version of figure 4c indicates less than 70%. Please, amend.

Reply: Agree. We have carefully checked this data and found that the percentage is 67%. Therefore, we have modified this sentence and write that "more than 65% of the HSC

population is found within a distance of 10 μ m from Caveolin-1+ AECs (Fig. 4c; Supplementary Fig. 5d)".

Reviewer #2 (Remarks to the Author):

The authors have addressed all of my concerns. Congratulations on an excellent manuscript.

Reply: We are most grateful for the positive comments and suggestions by the reviewer, which have helped us to improve the manuscript.